# Cryo-EM structures of the *E. coli* Ton and Tol motor complexes

Herve Celia [1], Istvan Botos [1], Rodolfo Ghirlando[1], Denis Duché [2], Bridgette M. Beach [1], Roland Lloubes[2] & Susan K. Buchanan [1] ✉

The Ton and Tol motor proteins use the proton gradient at the inner membrane of Gram-negative bacteria as an energy source. The generated force is transmitted through the periplasmic space to protein components associated with the outer membrane, either to maintain the outer membrane integrity for the Tol system, or to allow essential nutrients to enter the cell for Ton. We have solved the high-resolution structures of the *E. coli* TonB-ExbB-ExbD and TolA-TolQ-TolR complexes, revealing the inner membrane embedded engine parts of the Ton and Tol systems, and showing how TonB and TolA interact with the ExbBD and TolQR subcomplexes. Structural similarities between the two motor complexes suggest a common mechanism for the opening of the proton channel and the propagation of the proton motive force into movement of the TonB and TolA subunits. Because TonB and TolA bind at preferential ExbB or TolQ subunits, we propose a new mechanism of assembly of TonB and TolA with their respective ExbBD and TolQR subcomplexes and discuss its impact on the mechanism of action for the Ton and Tol systems.

The Ton and Tol systems are sophisticated molecular machineries essential for virulence and survival of Gram-negative bacteria. Both systems rely on the proton gradient at the inner membrane to produce force and movement that are transmitted to target proteins associated with the outer membrane. Ton is involved in the uptake of essential nutrients such as siderophore-iron, vitamin B12 and carbohydrates, while Tol maintains outer membrane integrity and plays an active role in cell division[1,2]. Ton and Tol are also the targets of bacteriocins and bacteriophages that bind to specific outer membrane receptors and hijack the Ton or Tol system to enter the periplasm and kill bacteria with high efficiency[3].

Ton drives active transport of nutrients across the outer membrane. In the inner membrane, TonB, ExbB and ExbD subunits form the TonB-ExbBD complex. Proton translocation through the ExbBD subcomplex generates movement that is transmitted via the pivotal TonB subunit to specific TonB Dependent Transporters (TBDTs) in the outer membrane. The C-terminal domain of TonB interacts with the nutrient loaded TBDT and upon activation, exerts force to open a channel through the TBDT, allowing nutrient uptake (Fig. 1)[4,5].

For Tol, the TolA, TolQ and TolR subunits form the TolAQR complex in the inner membrane. At the outer membrane, the lipoprotein Pal interacts with the soluble TolB protein. Upon energization, TolA displaces TolB from Pal, allowing Pal to interact with the peptidoglycan (PG) cell wall, stabilizing the connection between PG and the outer membrane (Fig. 1)[2].

Because Ton and Tol are essential for bacterial virulence and are targets of specific bacteriocins, it is important to understand how the different proteins involved interact with each other and how this network of interactions is utilized to allow proton translocation and force propagation.

Numerous structures of TBDTs have been reported, along with ExbBD subcomplexes and periplasmic fragments of TonB and ExbD[5,6]. Recent crystallographic studies revealed how TonB and ExbD interact in the periplasm[7,8].

[1]Laboratory of Molecular Biology, National Institute of Diabetes and Digestive and Kidney Diseases, National Institutes of Health, Bethesda, MD 20892, USA. [2]Laboratoire d'Ingénierie des Systèmes Macromoléculaires, UMR7255 CNRS/Aix-Marseille Université, Institut de Microbiologie de la Méditerranée, 13402 Marseille, France. ✉e-mail: susan.buchanan2@nih.gov

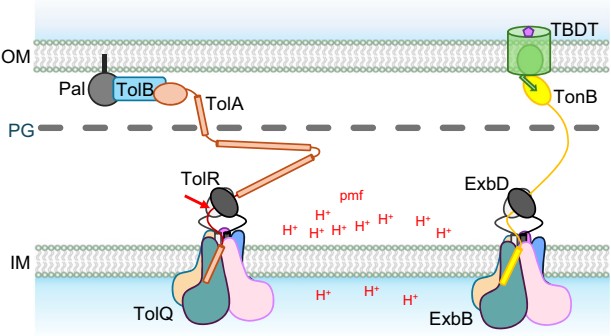

**Fig. 1 | The Ton and Tol systems.** Schematic representation of the Tol and Ton systems. Tol is represented on the left: five TolQ subunits assemble at the inner membrane (IM) to form a pentamer that defines a central hydrophobic pore in which a dimer of TolR subunits resides. The TolA subunit has one transmembrane (TM) domain, a very elongated periplasmic linker with predicted helical domains, and a folded C-terminal domain that crosses the peptidoglycan layer (PG) and interacts with the TolB-Pal complex at the outer membrane (OM). The proton gradient, or pmf, at the IM is used as an energy source by the TolQR subcomplex. The force generated is transmitted by TolA to displace TolB from Pal, allowing Pal to interact with the PG. The architecture of the Ton system, right, is similar: ExbB and ExbD are homologous to TolQ and TolR and have the same architecture. The elongated periplasmic linker of TonB crosses the PG layer and the C-terminal domain of TonB interacts with TonB Dependent Transporters (TBDTs) at the OM. The force generated by the proton translocation at the ExbBD subcomplex is transmitted by TonB to open a channel in the TBDT, allowing diffusion of the bound nutrient in the periplasmic space. The red arrow on TolA shows the location where the TEV site was introduced.

The stoichiometry of the ExbBD subcomplex has been a matter of debate as ExbB/ExbD ratios of 4/2, 5/1, 6/3 and 5/2 have been reported[9–12]. A four stage model has also been proposed for the Ton activity, in which the oligomeric ratio of ExbB, ExbD and TonB is different for each stage[13]. However, the 5/2 ratio is likely to correspond to the physiological state as it has also been found for *S. marcescens* ExbBD, *E. coli* TolQR, and the closely related MotAB complexes that use the pmf to drive the flagellar rotation[14–18].

For the Tol system, most of the structural knowledge comes from soluble fragments or complexes of periplasmic domains of Pal, TolB, TolR and TolA (see[2]). As mentioned above, a 4.3 Å resolution cryo-EM structure of *Ec*TolQR shows the five TolQ to two TolR stoichiometry, but the low resolution of the map did not allow structure building with high confidence[18].

The ExbBD and TolQR subcomplexes are highly homologous. TonB and TolA are less similar, reflecting their different functions, but do have in common the conserved SHLS motif (Ser-X$_3$-His-X$_6$-Leu-X$_3$-Ser) in their TM domain that is involved in the interaction of TonB with ExbBD and TolA with TolQR[19,20]. The two systems are similar to the point that they are able to complement each other, meaning that some Ton and Tol activities are still detected in *exbBD* or *tolQR* knockout strains, but absent in a double knockout[1]. This cross-complementation suggests that TonB can interact with TolQR and TolA with ExbBD, and that the use of the pmf and propagation of force to TonB and TolA are similar.

Our knowledge of how TonB or TolA interact with ExbBD or TolQR subcomplexes in the membrane is currently limited. One TonB-ExbBD structure was reported at 3.8 Å resolution[15]. The *P. savastanoi* cryo-EM map of TonB-ExbBD shows the five ExbB to two ExbD architecture, with one TonB transmembrane (TM) domain interacting with TM1 of one ExbB subunit. Unfortunately, because of low occupancy and/or local disorder, the local resolution was too low in this region of the map to allow a high resolution structure of TonB to be built.

Here, we report the cryoEM structures of *E. coli* TonB-ExbBD and TolAQR complexes at 2.8 Å and 3.0 Å resolution, respectively. One

TonB binds at the periphery of the ExbBD subcomplex through limited interactions with TM1 and a small N-terminal amphipathic helix α−1 of one ExbB subunit. The same set of interactions is found for TolA with TolQR, but two TolA subunits are bound per TolQR subcomplex. For both TonB-ExbBD and TolAQR, the structures highlight a network of conserved identical residues that connect the TM of TonB or TolA with the TM3 of ExbB or TolQ, respectively. These residues may be involved in signal transduction via TonB or TolA to open a proton channel in ExbBD or TolQR. We also found that TonB and TolA bind to preferential ExbB and TolQ subunits. We propose that the interaction of TonB with ExbBD, and TolA with TolQR, is modulated through additional interactions in the periplasm between TonB and ExbD, and TolA and TolR. Based on this high resolution structural information, we propose a new sequential model for the assembly of the TonB-ExbBD and TolAQR complexes and discuss its implications for the mechanistic models of the Ton and Tol systems.

## Results

### Structure of the *Ec* TolAQR complex

The *Ec*TolAQR complex was solubilized from membranes using the detergent Lauryl Maltose Neopentyl Glycol (LMNG) or Dodecyl Maltose Neopentyl Glycol (DMNG) and purified using a streptag-II on TolA. The Size Exclusion Chromatography (SEC) elution profile was broad (Suppl. Fig. 1A, B), and cryo-EM Single Particle Analysis (SPA) yielded structures at low resolution. Nevertheless, two-dimensional (2D) classification showed pentameric complexes that formed higher oligomers (Suppl. Fig. 1C). We hypothesized that the elongated periplasmic domain of TolA was the main source of heterogeneity, and we engineered a construct with a TEV proteolysis site between Ile50 and Asp51. This new construct copurified with TolQ and TolR and SEC yielded two elution peaks after TEV proteolysis (Suppl. Fig. 1A, B). The first SEC elution peak contained a mixture of dimers and monomers of the complex, but analysis of the dimer particles only yielded low resolution 3D structures. Cryo-EM images showed homogeneous particles of monomers for the second SEC elution peak (Suppl. Fig. 1D), which was used for high resolution cryoEM data collection and image analysis.

A total of 59,369 particles were used to calculate a three-dimensional (3D) map of the TolAQR complex in LMNG at 2.95 Å resolution (Fig. 2, Suppl. Fig. 2, Table 1). The structure reveals most of the TolQ subunit (residues 2-225), the N-terminus and TM of TolR (8-40) and the TM of TolA (4-34). The periplasmic domains of both TolA and TolR were not visible, suggesting these regions are flexible. TolAQR is a pentamer of TolQ, a dimer of TolR inside the TolQ pentamer, and two TM of TolA binding at the periphery of the pentamer (Fig. 2B, C). The complex dimensions are 100 × 80 x 80 Å, the bulk of the cytoplasmic domains of TolQ extend 60 Å into the cytoplasm, and the periplasmic regions protrude about 20 Å into the periplasm (Fig. 2B).

TolQ consists of α-helices connected by short loops. It has an elongated shape with dimensions 100 × 30 x 30 Å (Fig. 2A, D). The N-terminus resides in the periplasm. The four first residues are disordered and followed by a short amphipathic helix α−1 (residues 5-12) that lies parallel to the periplasmic leaflet of the membrane. A short loop (13-15) connects with elongated helix α−2 (16-57) that crosses the membrane and protrudes into the cytoplasm, followed by another loop (58-61), a short helix α−3 (62-72), loop (73-78), helix α−4 (79-94), loop (95-101) and helix α−5 (102-125) that points back toward the membrane. Helix α−6 (127-155) crosses the membrane, with a kink at residue Pro138, and protrudes into the periplasm. Helix α−6 is followed by a periplasmic loop (156-163) that connects with the elongated helix α−7 (164-225) that crosses the membrane and extends towards the cytoplasm. The TM domain of helices α−2, α−6 and α−7 have their main axis tilted 30 to 35° relative to the perpendicular axis to the membrane (Fig. 2A, D).

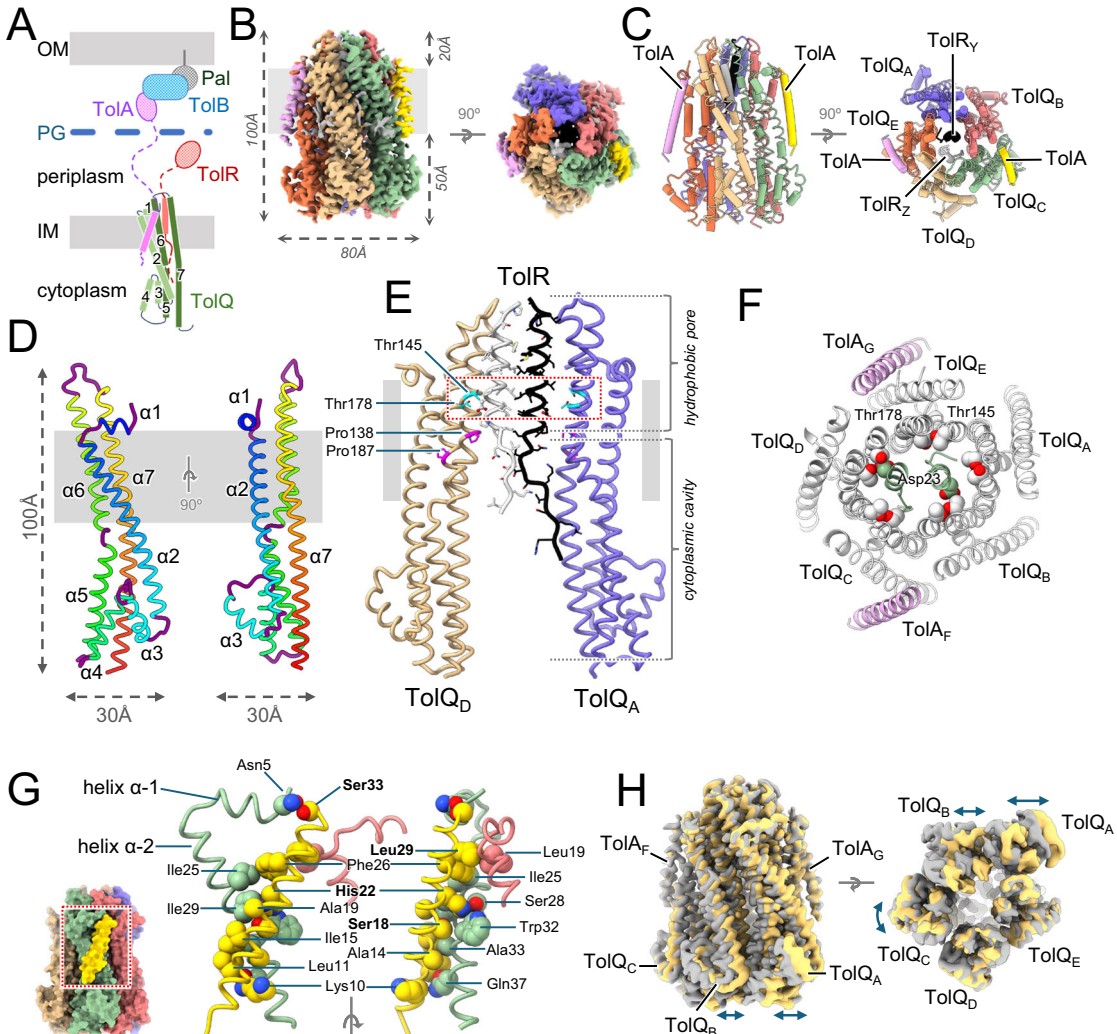

**Fig. 2 | Structure of TolAQR. A** Schematic drawing of the Tol system. TolA (purple) and TolR (red) have one TM domain at the IM, TolQ (green) has seven α-helices, and three TM domains. Periplasmic domains of TolA and TolR, not visible in the structure, are shown with dashed lines for the linkers and ovals for the C-terminal domains. TolA crosses the peptidoglycan layer (blue) and interacts with the TolB-Pal (blue and grey) complex anchored in the OM. **B** Cryo-EM structure of the *E. coli* TolAQR complex. TolAQR density map in LMNG (micelle masked), viewed perpendicular to the membrane (grey rectangle) plane (left), and to the periplasm (right). **C** Cartoon representation of TolAQR The labelling scheme follows PDB entry 6TYI[9]: TolQ_A blue, TolQ_B coral, TolQ_C green, TolQ_D tan, TolQ_E orange, TolR_Y black, TolR_Z light grey, TolA_F gold and TolA_G plum. **D** Cartoon representation of TolQ. Views from the membrane (grey rectangle) with α-helices in rainbow colors and loops in purple. **E** Cartoon representation of TolQ_{A,D} and TolR_{Y,Z}, showing the hydrophobic pore and cytoplasmic hydrophilic cavity from the membrane plane.

TolQ_{B,C,E} and TolA_{F,G} omitted for clarity. Conserved Pro138 and Pro187 (magenta stick) form a kink in helices α−6 and α−7. Side chains shown as stick, with conserved Asp23 (ball and stick) and Thr138 and Thr178 (cyan stick). **F** TolAQR cross-section in the membrane plane, observed from the periplasm. Cartoon representation of TolA_{FG} (plum), TolQ_{ABCDE} (light grey), and TolR_{YZ} (green). Asp23, Thr138 and 178 side chains shown as spheres. **G** Details of the interactions between TolA_F and TolQ_{BC}. For clarity only α−1 and TM of α−2 of TolQ_C, and part of α−1 and TM of α−2 of TolQ_B are shown, together with TolA_F. TolAQ shown in cartoon, contacting side chains (using PDBsum[61]) as spheres and SHLS motif residues identified with bold labels. **H** Structural variability of the TolAQR complex. Superimposition of density maps for TolA_{TEV}QR-I (grey) and TolA_{TEV}QR-II (yellow) observed from the membrane plane (left) or periplasm (right). Arrows show differences (see also the Suppl. Video).

The five TolQ subunits have a similar fold but exhibit variability in the orientations of helices α−6 and α−7 that line the hydrophobic pore. After fitted alignment and superposition of the five TolQs, TolQ chains B, C and D are similar, while the axis of helix α−6 of TolQ chain A (TolQ_A) is tilted toward α−2, and the axis of the α−7 TM of TolQ_E is tilted in the opposite direction (Suppl. Fig. 3A). The pivot points for these different conformations lie at conserved Pro138 on α−6 and Pro187 on α−7 (Fig. 2E, Suppl. Fig. 3A).

TolQ assembles as a pentamer. The monomers interact with one another along their α−5 and α−7 helices in the cytoplasm, and α−6 and α−7 in the membrane and periplasm. The inside of the pentamer defines an elongated pore that can be divided into two regions: a 60 Å long hydrophilic cavity that starts at the cytoplasmic opening and

protrudes into the membrane, followed by a 30 Å long hydrophobic pore that starts in the middle of the membrane and extends to the opening on the periplasmic side (Fig. 2E). Prolines 138 in α−6 and 187 in α−7 sit at the interface between the hydrophobic pore and hydrophilic cavity.

The structure of the TolR dimer was built from residues 12 to 37 for TolR_Z and 8 to 40 for TolR_Y. Each TolR comprises an unfolded region (8-18) stabilized through contacts with TolQ residues in the hydrophilic cavity, followed by an α-helix (19-40) that sits in the TolQ hydrophobic pore (Fig. 2E, Suppl. Fig. 3C). The N-terminal unfolded regions point in different directions in the hydrophilic cavity.

The two TolR α-helices are not at the same height in the pore, shifted about half a helical turn, with TolR_Y closer to the hydrophilic

**Table 1 | CryoEM data collection, structure determination and model statistics**

| | TonB-ExbBD | | | TolA_TEV_QR | | TolAQR |
|---|---|---|---|---|---|---|
| **Data collection** | | | | | | |
| Nominal magnification | 105,000x | | | 105,000x | | 45,000x |
| Voltage (kV) | 300 | | | 300 | | 200 |
| Exposure time (s/frame) | 0.05 | | | 0.075 | | 0.08 |
| Number of frames | 46 | | | 30 | | 30 |
| Total dose (ē/Å²) | 69.9 | | | 61.4 | | 69.9 |
| Defocus range (µm) | -0.8 to -2.8 | | | -0.8 to -2.8 | | -0.7 to -2.5 |
| Pixel size (Å) | 0.83 | | | 0.83 | | 0.89 |
| **Image processing** | | | | | | |
| Micrographs selected | 5,784 | | | 7,811 | | 5,084 |
| Initial particle images (no.) | 5,414,663 | | | 11,112,982 | | 1,519,049 |
| | TonB bound to ExbB_C | TonB bound to ExbB_E | TonB bound to ExbB_A | TolA_TEV_QR-I | TolA_TEV_QR-II | |
| Final particle images (no.) | 227,584 | 30,556 | 40,906 | 59,369 | 78,674 | 107,631 |
| Symmetry imposed | C1 | C1 | C1 | C1 | C1 | C1 |
| FSC threshold | 0.143 | 0.143 | 0.143 | 0.143 | 0.143 | 0.143 |
| Final map resolution (Å) | 2.80 | 3.16 | 3.19 | 2.94 | 3.18 | 5.40 |
| Resolution range (Å) | 2.7-3.5 | 2.9-4.5 | 3.0-4.4 | 2.7-3.9 | 2.9-4.5 | |
| **Atomic model** | | | | | | |
| Number of protein residues | 1209 | 1199 | 1199 | 1221 | 1206 | |
| **Validation** | | | | | | |
| Most favored (%) | 99.50 | 99.07 | 99.24 | 99.25 | 98.57 | |
| Allowed (%) | 0.50 | 0.93 | 0.76 | 0.75 | 1.43 | |
| Disallowed (%) | 0 | 0 | 0 | 0 | 0 | |
| Rotamer outliers (%) | 0.53 | 0.43 | 0.43 | 0.30 | 0.51 | |
| r.m.s.d Bond lengths (Å) | 0.011 | 0.011 | 0.011 | 0.012 | 0.012 | |
| r.m.s.d Bond angles (°) | 1.626 | 1.624 | 1.603 | 1.628 | 1.639 | |
| Clashscore | 0.48 | 2.40 | 1.26 | 1.91 | 2.03 | |
| Map CC (mask) | 0.88 | 0.88 | 0.88 | 0.86 | 0.84 | |
| Map CC (volume) | 0.88 | 0.88 | 0.88 | 0.85 | 0.83 | |
| **Deposition ID** | | | | | | |
| PDB ID | 9DDO | 9DDP | 9DDQ | 9DDM | 9DDN | |
| EMDB ID | EMD-46778 | EMD-46779 | EMD-46780 | EMD-46776 | EMD-46777 | EMD-70142 |

cavity than TolR_Z. The essential TolR Asp23 residues are close to the conserved TolQ threonines 145 and 178 but have different environments (Fig. 2E, F). These threonines form a polar ring in the hydrophobic pore (Fig. 2F). Asp23 on TolR_Z contacts threonines on TolQ_D, while Asp23 on TolR_Y points toward the hydrophilic cavity and is too distant to interact with either threonine on TolQ_A or TolQ_B (Fig. 2E, F).

Residues 4 to 34 of TolA (ATEQNDKLKRAIII**S**AVL**H**VILFAA-L**IWSS**F), which include the conserved SHLS motif[20], were built (Fig. 2B, C, and G, Suppl. Fig. 3D). TolA TM is a 40 Å long, straight α-helix, tilted 20° relative to the perpendicular axis to the membrane plane, in the opposite direction of TolQ α−2 TM (Fig. 2A, G). The two TolAs, labelled TolA_F and TolA_G, have a similar fold (Suppl. Fig. 3D), and share a similar surface of interactions with TolQ (Table 2). The highly conserved and essential TolA His22[21] makes multiple contacts with TolQ Ser28, Trp32 and Ile29 (Fig. 2G). A detailed list of interactions between TolA and TolQ are summarized in Suppl. Fig. 4.

The stoichiometry of the TolA_TEV_QR complex is two TolA, five TolQ and two TolR. The structure was obtained with a truncated version of TolA containing a TEV sequence inserted between residues Ile50 and Asp51, which lies in the conserved motif Ile50-Asp-Ala-Val-Met-Val-Asp56 (IDAVMVD) that is predicted to interact with TolR[7]. This modification could affect the assembly and/or activity of the TolAQR complex.

A cryo-EM dataset of native, full-length TolAQR in DMNG was collected at 200 keV and SPA yielded a 5.4 Å resolution structure calculated with 107,631 particles (Suppl. Fig. 5A, Table 1). This complex does not have the TEV sequence inserted in TolA and therefore has the intact R-box motif. A perfect fit was found between this map and the TolA_TEV_QR structure, showing that insertion of the TEV site on TolA did not affect the stoichiometry of the complex (Suppl. Fig. 5B).

Furthermore, the insertion of the TEV sequence had no significant effect on Tol activities. Using a *ΔtolA* strain, we found that complementation with the tolA_TEV construct or wt tolA restored all Tol dependent activities, i.e. ability to grow on SDS containing medium, sensitivity to the Tol dependent colicin A, and absence of filamentous cells during cell division (Suppl. Fig. 6).

The TolA_TEV_QR complex is highly dynamic, especially in the cytoplasmic region. The final 3D classification step in Relion4[22] yielded three distinct classes (Suppl. Fig. 2). The class that showed the best resolution was chosen to calculate the structure described above. Another 3D class that comprised 78,674 particles showed significant displacements of some of the TolQ subunits (Fig. 2H). Further refinement yielded a 3.2 Å resolution 3D density and a second structure of the TolA_TEV_QR complex, named TolA_TEV_QR-II was built (Table 1). Comparison of the two structures shows significant displacement of the cytoplasmic regions of TolQ chains A, B and C, whereas the structures of the transmembrane and periplasmic regions are virtually

**Table 2 | Interface buried surface areas (A²)**

|  | TonB-ExbBD |  |
|---|---|---|
| ExbB$_{D-C}$ | 1551.2 |  |
| ExbB$_{B-A}$ | 1529.0 |  |
| ExbB$_{E-A}$ | 1485.4 |  |
| ExbB$_{C-B}$ | 1392.0 |  |
| ExbB$_{E-D}$ | 1378.4 |  |
| TonB-ExbB$_C$ | 477.6 |  |
|  | TolA$_{TEV}$QR-I | TolA$_{TEV}$QR-II |
| TolQ$_{D-C}$ | 1041.9 | 981.5 |
| TolQ$_{B-A}$ | 1092.1 | 1053.9 |
| TolQ$_{E-A}$ | 1254.3 | 1275.3 |
| TolQ$_{C-B}$ | 1381.1 | 1368.1 |
| TolQ$_{E-D}$ | 984.1 | 907.8 |
| TolA$_F$-TolQ$_C$ | 624.8 | 472.3 |
| TolA$_F$-TolQ$_B$ | 135.5 | 110.2 |
| TolA$_F$-TolQ$_{CB}$ | 760.3 | 582.5 |
| TolA$_G$-TolQ$_E$ | 599.1 | 606.1 |
| TolA$_G$-TolQ$_D$ | 196.7 | 180.1 |
| TolA$_G$-TolQ$_{ED}$ | 795.8 | 786.2 |

Interfaces calculated with PISA[60]

identical. A morph movie reveals that the cytoplasmic regions of TolQ$_A$ and TolQ$_B$ sway together laterally with up to 9 Å displacements relative to the membrane plane, and the TolQ$_C$ cytoplasmic region rotates along its elongated axis perpendicular to the membrane (Suppl. Movie). The N-terminal region of TolR$_Y$ in the hydrophilic cavity also shows two distinct conformations, with a rearrangement of the Leu13-Ser14-Glu15 region (Suppl. Fig. 3C, Suppl. Movie).

**Structure of the *Ec* TonB-ExbBD complex**
The *Ec*TonB-ExbBD complex was solubilized using the detergent n-Dodecyl-β-D-Maltopyranoside (DDM). Using DDM throughout the purification procedure led to significant loss of TonB at each step, suggesting that the interaction between TonB and the ExbBD sub-complex was labile. To stabilize TonB-ExbBD, DDM was replaced with amphipol PMAL-C12 after the first IMAC affinity step, and the remaining purification was performed in the absence of detergent. Using these conditions, the TonB-ExbBD complex was stable and showed no sign of degradation over extended periods of storage at 4 °C (Suppl. Fig. 7A, B).

SPA of *Ec*TonB-ExbBD in PMAL-C12 yielded a 2.8 Å resolution consensus 3D map using 227,384 particles (Fig. 3B, Suppl. Fig. 8, Table 1). The structure is similar to the reported 3.3 Å resolution structure of ExbBD in MSP1D1 nanodiscs[9], with the addition of a TM helix of one TonB bound to ExbB chain C (Fig. 3B, C, and E, chain IDs according to PDB 6TYI[9]). The stoichiometry of the complex is one TonB subunit, five ExbB and two ExbD. This ratio was confirmed by Mass Photometry (MP) and Sedimentation Velocity Analytical Ultracentrifugation (SV-AUC) experiments performed using TonB-ExbBD and ExbBD in PMAL-C12 (Suppl. Fig. 7C, D).

As previously described[9,10], ExbB consists of 7 α-helices connected with short loops (Fig. 3A, Suppl. Fig. 3B). The TM domains of helices α−2, α−6 and α−7 are tilted 30° to 40° relative to the perpendicular axis of the membrane plane, and five ExbB subunits assemble around a pseudo-5-fold symmetry axis with the α−6 helix and TM region of α−7 defining a hydrophobic pore that extends 10 Å toward the periplasm. A dimer of ExbD TM domains (residues 21-42) dock into the pentameric ExbB hydrophobic pore. The two ExbD TM helices are parallel but shifted relative to one another by about half a helical turn (Fig. 3D). Inside the cytoplasmic cavity formed by the pentamer of ExbB,

densities for the N-terminal part of ExbD allowed us to build a model starting at residue Asp11. The fold of these two regions of ExbD is similar but their orientations relative to their respective TM domains are different, resulting in an asymmetrical dimer. ExbD Met15 and Glu14 residues are stabilized through H-bonds with ExbB$_{AD}$ Asn196 and Arg200 (Fig. 3D).

Clear densities in the membrane region allowed us to build a structure for TonB residues 10 to 32 (PWPTLL**S**VC**IH**GAVVAG**L**-LYT**S**GV), which corresponds to the TM helix region with the SHLS motif (Fig. 3E, Suppl. Fig. 3E). The TonB TM is a 35 Å long α-helix, tilted about 15° from the perpendicular axis of the membrane, and interacts with the ExbB$_C$ α−1 helix and α−2 TM (Fig. 3B, C, and E). The axes of the TonB and ExbB α−2 TM helices are tilted in opposite directions.

Most of the contacts between TonB and ExbB$_C$ are located within a region close to the cytoplasm. In this region, the highly conserved TonB residues Ser16 and His20 interact with the ExbB highly conserved Trp38 and Ser34 (Fig. 3E). The interface area between TonB and ExbB$_C$ is 478Å², involving the interaction between ten residues on ExbB$_C$ with seven on TonB (Fig. 3E, Table 2, Suppl. Fig. 4).

The cryo-EM map also shows elongated densities in the membrane region at the interface between adjacent ExbB subunits. These densities were attributed to lipids that copurified with the TonB-ExbBD complex. The resolution was high enough to build two phosphatidy-lethanolamine (PE) lipid molecules (Suppl. Data Fig. 10A). These two PE are located at the interface between ExbB chains A and B, and D and E. No external lipids were added during purification. As observed for *Sm* ExbB (pdb 6YE4[14]), these tightly bound lipids are anchored through interaction between their polar head group and an arginine residue (Arg200 for *Ec* ExbB, Arg237 for *Sm* ExbB).

While exploring the optimal conditions to determine the structure of TonB-ExbBD, several cryo-EM datasets were collected at 200 keV and analyzed. Among these, three independent analyses yielded structures at resolutions close to 4 Å, which were fitted and oriented with the ExbBD structure (PDB 6TYI[9]). The three maps showed an extra elongated density in the membrane region that was attributed to the TM of TonB, but at different positions: two showed TonB interacting with ExbB$_E$, and one with ExbB$_C$.

TonB interacts with ExbB$_C$ in the consensus map. To eventually detect structural heterogeneities and/or TonB binding at different ExbB, we performed an additional 3D classification step with the 227,384 particles that yielded the 2.8 Å resolution map. The 3D classification was performed within Relion4[22], using the local angular search and fine angular step options (Suppl. Fig. 9). Inspection of the five resulting classes showed that three classes (69% of the particles) matched the structure of the consensus map. The two others, representing 18% and 13% of the particles, clearly showed two different orientations of the ExbD dimer (Fig. 3F, Suppl. Fig. 9). These two maps were further oriented and fitted with the ExbBD structure[9]. Each of the two maps resulted in a better fit with ExbBD after rotation along the pseudo-5-fold and alignment of the ExbD regions. In these new orientations, TonB was found to bind ExbB$_E$ (18% of the particles) or ExbB$_A$ (13% of the particles).

The 2.8 Å resolution consensus structure thus corresponds to a mixed population of TonB-ExbBD, aligned on the TonB subunit, but with ExbBD in three different orientations along the pseudo-5-fold axis. Because about 70% of the particles are correctly aligned, the contribution of the ~30% misaligned particles did not affect the model building of ExbB and ExbD, but merely contributed to the quality of the consensus map for TonB. A structure was built for these two maps (Fig. 3G, Table 1). Except for the binding of TonB to different ExbB, the three structures do not show any noticeable structural difference for the ExbBD subcomplex, nor in the way TonB interacts with ExbB.

For the remainder of this report, the designation TonB-ExbBD will refer to the consensus 2.8 Å resolution structure with TonB bound to ExbB chain C.

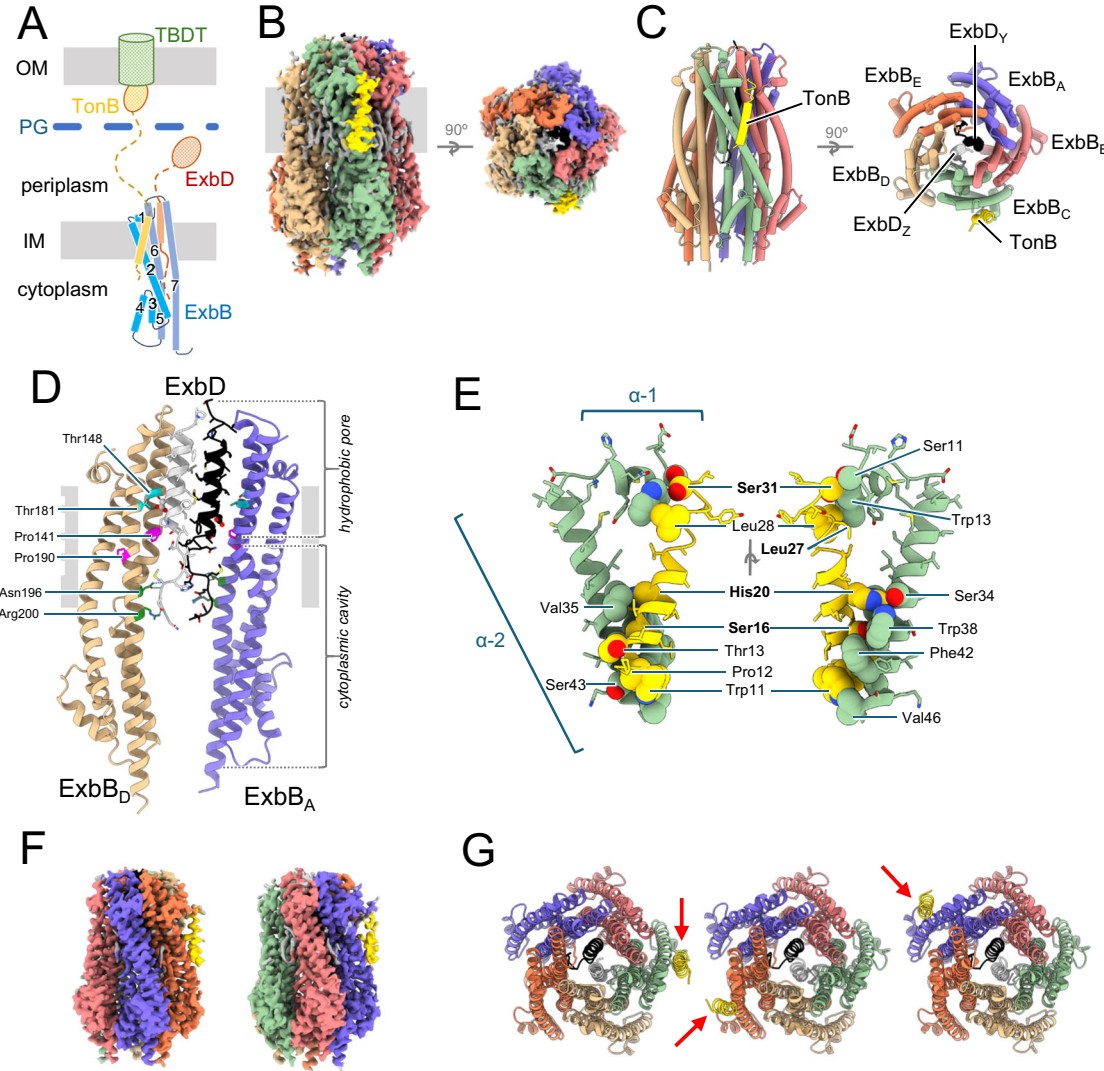

**Fig. 3 | Structure of *Ec*TonB-ExbBD. A** Schematic drawing of the Ton system. TonB (yellow) and ExbD (red) have one TM domain at the IM, ExbB (blue) has seven α-helices, and three TM domains. Periplasmic domains of TonB and ExbD, not visible in the structure, shown with dashed lines for linkers and ovals for C-terminal domains. TonB crosses the PG (blue) and interacts with the TBDT transporter (green) in the OM. **B** Cryo-EM structure of the *E. coli* TonB-ExbBD complex. TonB-ExbBD isosurface density map in PMAL-C12 (micelle masked), observed perpendicular to the membrane (grey rectangle, left), and from the periplasm (right). Lipids are colored grey. **C** Cartoon representation of TonB-ExbBD, viewed as in (**B**). The labelling scheme follows PDB entry 6TYI[9]. ExbB$_A$ blue, ExbB$_B$ coral, ExbB$_C$ green, ExbB$_D$ tan, ExbB$_E$ orange, ExbD$_Y$ black, ExbD$_Z$ light grey, TonB$_F$ in gold. **D** Cartoon representation of ExbB$_{A,D}$ and ExbD$_{Y,Z}$, showing the hydrophobic pore and cytoplasmic hydrophilic cavity viewed from the membrane (grey rectangle). For clarity, ExbB$_{B,C,E}$ and TonB$_F$ were omitted. The ExbB α−5, α−6 and α−7 helices

interact in the pentamer to form the hydrophilic cytoplasmic cavity and the hydrophobic pore, in which the ExbD TM domains reside. Conserved Pro141 and Pro190 (magenta stick) form a kink in helices α−6 and α−7. The two essential Asp23 side chains (ball and stick), and the conserved Thr148 and Thr181 (cyan stick), are shown. Conserved Asn196 and Arg200 on ExbB$_{A,D}$ (green stick) form H-bonds (green dashed lines) with Met15 and Glu14 on ExbD$_{Y,Z}$. **E** TonB and ExbB$_C$ interaction details in cartoon representation. For clarity only α−1 and TM of α−2 of ExbB$_C$ are shown, together with TonB. Contacting side chains (with PDBsum[61]) shown as spheres and labelled. TonB residues from the SHLS motif are labeled bold. **F** 3D density maps of TonB-ExbBD with TonB interacting with ExbB$_E$ (left) or ExbB$_A$ (right). **G** Cartoon representation of TonB-ExbBD structures, viewed from the periplasm, with TonB (gold) interacting with ExbB$_C$ (green) (left), ExbB$_E$ (tan) (center) or ExbB$_A$ (blue) (right). TonB subunits are shown with red arrows, ExbBD subcomplexes are in the same orientation.

## Discussion

The high-resolution structures of both TonB-ExbBD and TolAQR complexes provide us with the unique opportunity to determine, at the molecular level, the structural elements shared by the two systems, and how the TonB and TolA subunits might modulate the gating of the proton channel.

ConSurf [23] (https://consurf.tau.ac.il) was used to determine and visualize the degree of conservation of each residue in our structures. As previously reported[24], most of the highly conserved residues are localized within the hydrophobic pore, where the proton translocation is predicted to occur (Fig. 4, Suppl. Fig. 11). The respective local environments of the two essential Asp residues on the TM domains of

either ExbD or TolR are highly similar (Fig. 2E, F, Fig. 3D). The same configuration is found in the cryo-EM structures of the MotAB and PomAB stators that use the proton or sodium gradient, respectively, to drive flagellar rotation (Suppl. Fig. 12). As for the ExbBD and TolQR subcomplexes, MotAB and PomAB share the 5 to 2 stoichiometry with a central hydrophobic pore shifted toward the periplasm that encompasses the dimer of MotB or PomB TM domains. The two essential Asp on MotB and PomB have the same asymmetrical environment, either in contact with a conserved Thr of MotA/PomA or pointing toward the cytoplasmic cavity (Suppl. Fig. 12)[25,26]. These striking structural similarities indicate that these four systems share a common ancestor[27]. They also suggest that the path and mechanism of

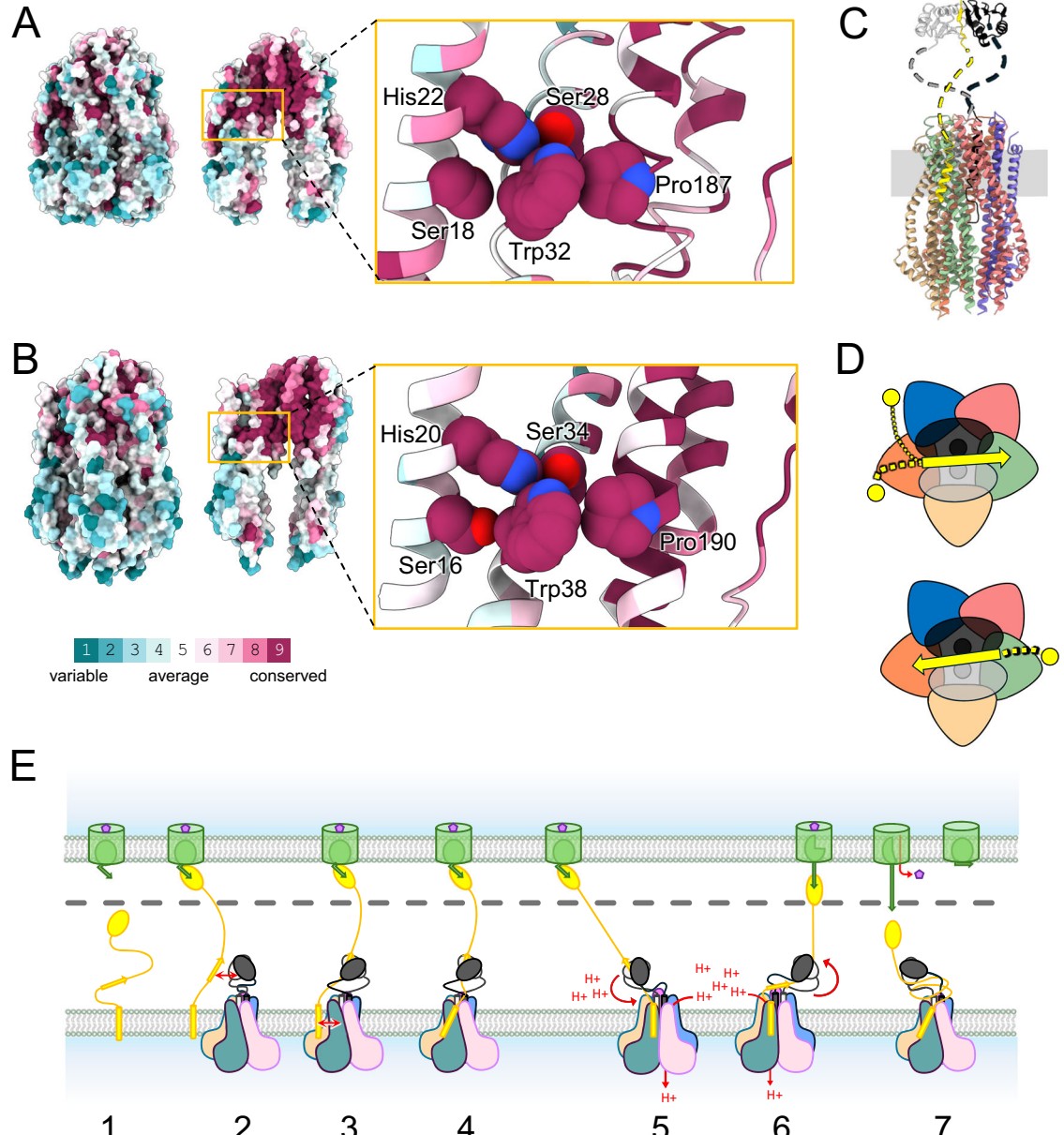

**Fig. 4 | Conserved network of residues connecting TolA/TonB and TolQ/ExbB TM3, preferential binding of TolA and TonB, and schematic model of Ton assembly and mechanism. A** Left panel: atomic surface representation of TolA$_{TEV}$QR highlighting the conservation of residues (Consurf [23]). Middle panel: TolQ chains (**A−D**) omitted to show the hydrophobic pore and TolR. Network of conserved residues connecting the TM of TolA with the TolQ TM3 is highlighted. Right panel: zoom in of the highlighted region, in cartoon representation. Residues that form the network are shown with spheres. **B** Same representation as in panel A for TonB-ExbBD. **C** Predicted structure of the full TonB-ExbBD complex in cartoon representation. Unfolded linkers are dashed lines. The different subunits are colored as in Fig. 3 with the periplasmic domains of TonB in gold, ExbD$_Y$ black and ExbD$_Z$ light grey. **D** Two schematic representations of the TonB-ExbBD complex, seen from the periplasm. ExbB subunits shown as rounded triangles, TonB (yellow) and ExbD (black and light grey) TMs as circles, ExbD periplasmic domains as ovals, the TonB linker connecting the TM and the D-box with yellow dashed lines, and the D-box as a yellow arrow. The distal C-terminal domain of TonB is not represented. The color code is the same as in (**C**). The upper panel has the D-box bound to the ExbD periplasmic dimer in one orientation, leading to the binding of the TonB TM domain either to ExbB$_E$ (orange) or ExbB$_A$ (blue). In the lower panel the D-box is in

the opposite direction, and the TonB TM domain binds ExbB$_C$ (green). **E** Hypothetical assembly and mechanistic model for Ton. (1) Monomeric TonB (yellow) diffuses in the cytoplasmic membrane. TBDT (green) in the OM binds a nutrient (purple), inducing the TonB box (green arrow) to extend into the periplasm. (2) TonB C-terminal domain binds the TBDT TonB box, and the TonB D-box (yellow arrow) comes close to the ExbD periplasmic dimer (grey and black). (3) TonB D-box binds the ExbD dimer and (4) TonB TM domain interacts with ExbB in the membrane (red arrow) forming a full TBDT-TonB-ExbBD complex. (5) TonB-ExbBD lateral diffusion in the membrane pulls on the TonB periplasmic domain and reorients the TonB TM domain. This change is transmitted to the ExbB TM3 and opens the pentameric pore. Proton translocation through the pore causes the ExbD TM dimer to rotate. (6) ExbD rotation wraps around the TonB periplasmic linker, exerting a pulling force on the TonB C-terminal domain towards the periplasm. (7) The pulling force gradually unfolds the TBDT plug domain and opens a channel allowing the nutrient to diffuse into the periplasm. The force necessary to further unfold the plug domain is greater than the pulling force and TonB detaches from the TonB box, dissipating the tension on the periplasmic linker. TonB-ExbBD reverts to the ground state with proton channel closed, and the TBDT plug domain refolds into the barrel.

proton or cation translocation are similar for the four systems, likely resulting in the rotation of the dimer of MotB/PomB/ExbD/TolR inside the pentameric pore as hypothesized for Mot[15,17].

In the Mot system, the MotB plug domain binds to MotA and maintains the proton channel in its closed state. Upon association with the flagellum, the MotB plug domain dissociates from MotA, allowing MotB C-terminal periplasmic domains to interact with the PG layer, and the proton channel to open[28]. The same gating mechanism is observed for the Pom system[29].

In Ton and Tol, there is no plug domain on the ExbD and TolR subunits, therefore the gating mechanism must be different. The opening of the proton channel is likely dependent on the association of TonB with ExbBD or TolA with TolQR, together with the signaling of TonB interacting with a nutrient loaded TBDT or TolA with TolB-Pal at the outer membrane. Comparison of the structure of $Ec$ExbBD[9] with $Ec$TonB-ExbBD shows no significant conformational change for ExbBD, suggesting that the association of TonB to ExbBD is not sufficient to trigger the opening of the proton channel. For Tol, we attempted to compare the structures of the TolAQR complex with the recently reported 4.2 Å resolution structure of TolQR[18]. While we see some noticeable differences between TolAQR and TolQR, especially for the cytoplasmic domains of TolQ, it is not clear whether these differences are due to the presence of TolA, or to the low resolution of the TolQR structure.

Upon opening of the proton channel, one of the two conserved Asp residues on ExbD or TolR likely translocates a proton through transient protonation of its carboxyl group and generates a power stroke. Based on the MotAB and ExbBD structures, different rotary models have been proposed[15,17,30]. These mechanistic models differ in the detailed sequence of molecular events that lead to the power stroke, but they all hypothesize that the dimer of MotB/ExbD TM domain rotates 36° for each proton translocated, and that the opening and closing of the proton channel alternates between the two essential Asp residues.

In the TonB-ExbBD and TolAQR structures the TM domains of ExbD or TolR are tucked in the ExbB or TolQ pentameric pores. In this tight configuration, it is hard to conceive that ExbD or TolR could rotate unless some or all the ExbB or TolQ subunits modify their conformations to widen the lumen of the pentameric pore. The ExbBD complex has been observed in different oligomeric states, ranging from 5 ExbB to 2 ExbD, 5 to 1, or 6 to 3[9–11]. While the 5 to 2 ratio likely represents the physiological state, it shows that ExbB has enough structural plasticity to form different oligomers and accommodate different ratios of ExbD subunits. The two structures of TolAQR described in this report show that TolQ can adopt different conformations, with a high degree of flexibility between the cytoplasmic and TM domains. This region has two conserved proline residues, Pro138 and Pro187, that introduce a kink in their respective α−6 and α−7 helices. These two prolines have equivalent residues in ExbB, MotA and PomA and have been found to be important for activity, especially for the proline in α−7 for ExbB ($Ec$ExbB Pro190), TolQ ($Ec$TolA Pro187), and in TM4 for MotA ($Ec$MotA Pro222)[31–34]. Prolines are known to form kinks in TM helices and to be involved in signal transduction[35,36]. These two prolines may modulate the widening of the hydrophobic pore that would not only loosen the restraints on the ExbD/TolR TM dimer, allowing it to rotate, but could also participate in the gating of the proton channel.

The distribution of conserved residues in our TonB-ExbBD and TolAQR structures shows that the most conserved regions are in the hydrophobic pore, while residues exposed to solvent and/or lipids are more variable (Fig. 4A, B). It also highlights a network of highly conserved residues that connect TonB/TolA with the TM3 of ExbB/TolQ. Ser16 and His20 on TonB, part of the SHLS motif, make multiple contacts with Ser34 and Trp38 on ExbB TM1, while Trp38 is also in contact with Pro190 on ExbB TM3 (Fig. 4B). The same arrangement is found for TolAQR, with Ser18 and His22 on $TolA_{F,G}$ and Ser28, Trp32 and Pro138 on $TolQ_{C,E}$ forming the same 3D network observed for TonB-ExbBD (Fig. 4A). In this configuration, a change of orientation of the TM of TonB or TolA could modify the conformation of $ExbB_C$ Pro190 or $TolQ_{C,E}$ Pro187 via the two conserved Ser and Trp residues on TM1. Most of these residues are critical as their mutations result in the inhibition or a complete loss of pmf dependent Ton and Tol activities (Suppl. Table 1).

It has been hypothesized that the signaling of TonB bound to a nutrient loaded TBDT involves tension on the TonB periplasmic linker generated by the tethering of TonB to the OM coupled with the diffusion of the TonB-ExbBD complex in the inner membrane[30,37]. This tension could affect the orientation of the TonB TM domain, and the associated conformational change of His20 and Ser16 would be transmitted to Pro190 on ExbB via Ser34 and Trp38. The conformational change of Pro190 could modify the orientation of ExbB α−7 helix, eventually widening the lumen of the hydrophobic pore. A similar mechanism would apply for the Tol system.

As deduced from the TonB-ExbBD structures, one TonB subunit associates with ExbBD via ExbB chains C, E or A (Fig. 3F, G). There are two TolA in the TolAQR complex, which are bound to TolQ chains C and E (Fig. 2). Further inspection of the non-sharpened $TolA_{TEV}QR$ 3D density map shows a stretch of weak densities close to TolQ chain A, suggesting that TolA can also bind $TolQ_A$ (Suppl. Fig. 10D). TonB and TolA thus bind to the same ExbB and TolQ chains C, E or A, but not to chains B and D. The preferential binding could be a consequence of the asymmetry of the ExbB and TolQ pentamers, induced by the presence of the ExbD and TolR dimers in the pore. The asymmetry within the pentamers is illustrated by the different interaction surfaces of the ExbB or TolQ subunits (Table 2). However, the regions of ExbB or TolQ involved in the interaction with TonB or TolA are equally exposed in the membrane and share the same conformation. Another explanation would be that the preferential binding is a consequence of other interactions within the TonB-ExbBD or TolAQR complexes.

For the Ton system, interactions of TonB and ExbD in the periplasm have been structurally characterized. Crystallographic structures of a soluble construct of $Ec$ExbD in complex with a peptide of $Ec$TonB have been reported[7,8]. The ExbD periplasmic domain forms a symmetric homodimer that binds to a conserved sequence on TonB (Suppl. Fig. 13A). Upon association with ExbD, the so-called D-box on $Ec$TonB (45-ISVTMVT-51) forms a β-strand that binds to β5-strands on each ExbD. The D-box binding site on ExbD exhibits 2-fold symmetry, meaning that the D-box can bind the ExbD dimer in either direction. The periplasmic domain of TolR has been shown to form at least two different dimers, with one similar to the ExbD dimer[8,38,39] (Suppl. Fig. 13C), and TolA, like TonB, has a conserved motif in its periplasmic domain that is suspected to act as a R-box motif[7] (Suppl. Fig. 11).

While the ExbD periplasmic dimer and the TonB D-box are not visible in the TonB-ExbBD maps, the full complex is likely organized as shown in Fig. 4C, with the periplasmic linkers of TonB and ExbD unfolded and dynamic (see also Supplementary Fig. 16 in Zinke et al.[8]). The binding of the D-box to ExbD is likely responsible for the preferential binding of TonB with ExbB. If the ExbD periplasmic dimer is not randomly oriented but restricted by the orientation of the ExbD TM dimer within the pore, the TonB D-box will only have limited access to ExbD and hence the TonB TM to ExbBD in the membrane.

It has been shown that a short motif in the ExbD periplasmic linker, the N-terminal Intermolecular Beta-Strand or NIBS, can stabilize a closed form of the dimer through self-association to form an intermolecular β-sheet[8] (Suppl. Fig. 13B). The closed state is in equilibrium with an open state in which the NIBS is unfolded. The periplasmic dimer is likely dynamic in both the closed and open states as it is not visible in the reported 3D maps of the ExbBD subcomplex[9,14]. Nevertheless, the closed state ought to further restrain the respective orientations of the ExbD periplasmic and TM dimers.

We hypothesize that the axes of the ExbD periplasmic and TM dimers are aligned as shown in Fig. 4D. In this configuration, the association of the TonB D-box on either side of the ExbD dimer would bring the TonB TM closer to ExbB chains C or E, then to chain A, but far from chains B and D, explaining why TonB is mostly found to bind ExbB chains C or E. It also suggests that the association of TonB to ExbBD is dictated first by contacts between TonB and ExbD in the periplasm, and then between the TM of TonB and ExbB in the membrane. Because of the extensive similarities between the Ton and Tol systems, i.e. the presence of a D- and R-box on TonB and TolA, the presence of the NIBS and equivalent dimer for ExbD and TolR, and the preferential binding of TonB or TolA to equivalent ExbB or TolQ subunits, a similar mechanism of assembly is likely to take place for Tol. An updated schematic model for the Ton assembly and usage of the proton motive force is summarized in Fig. 4E.

The main difference between the TonB-ExbBD and TolAQR complexes is the respective stoichiometries of TonB and TolA. The physiological significance of two TolA subunits in TolAQR is not clear. The binding of TolA to TolQ is more stable than TonB with ExbB, with an average buried surface area of 720 Å$^2$, compared to 480 Å$^2$ for TonB and ExbB. The overexpression of TolA, TolQ and TolR for structural analyses might result in an artefactual binding of multiple TolAs to TolQR. Cryo-EM observation of intact TolAQR solubilized complexes show that TolAQR can form dimers and higher oligomers, with most of them showing a similar orientation in the membrane. On the other hand, TEV digested TolA$_{TEV}$QR is mostly monomeric (Suppl. Fig. 1D, E), suggesting that interactions between the TolA periplasmic domains might influence the oligomeric state of TolAQR.

Both TolA and TonB have a conserved SHLS (Ser-3X-His-6X-Leu-3X-Ser) motif which is lined up on the same side of the TM helix (Suppl. Fig. 3D, E). The first Ser-His residues are the most structurally conserved and share the same network of interactions with the conserved Ser, Trp and Pro on both TolQ and ExbB as shown on Fig. 4A, B. This illustrates the importance of the His residue, shown to be essential for Tol or Ton activities[20,21]. The other Leu and Ser of the SHLS motif are not as well conserved for Ton (Suppl. Fig. 11C). The Ser-His residues of the SHLS motif might be important for the interaction between TolA/TonB and TolQ/ExbB and signal transduction, while Leu-Ser might play a role when the complexes are activated to generate force.

In conclusion, the TolAQR and TonB-ExbBD complexes show extensive structural similarities. Both share a network of conserved residues that connect the Ser-His of the SHLS motif on TolA and TonB to the TM1 and TM3 of TolQ and ExbB, which we hypothesize is involved in signal transduction. The observed preferential binding of TolA or TonB to equivalent chains of TolQ or ExbB is likely the consequence of interaction between TolA or TonB with TolR or ExbD in the periplasm, suggesting that the interactions in the periplasm and in the membrane are synchronized. The association of TolA to TolQR or TonB to ExbBD is not sufficient to trigger opening of the proton channel, therefore additional molecular events are needed to open both the proton channel and the hydrophobic pore.

## Methods

### Bacterial strains and plasmids
The bacterial strains and plasmids used in this study are listed in data Table 3.

*EctolA* was subcloned into pCDFDuet-1 (Novagen) with an N-terminal streptag-II using the Restriction-Free cloning technique[40]. The mutant of *EctolA* with a TEV site insertion was prepared by site-directed mutagenesis using the Q5 Site-Directed Mutagenesis Kit (New England Biolabs). Following TEV digestion, the sequence of the resulting N-terminal fragment is MGS<u>WSHPQFEKGSSKA</u>-TEQNDKLKRAIIISAVLHVILFAALIWSSFDENIEASAGGGGGSS<u>IENLYFQ</u> (streptag-II and TEV sequences are underlined). *EctonB* was subcloned into a pACYCDuet-1 vector with a C-terminal TEV protease site tag

### Table 3 | plasmids and strains

| **plasmid** | | |
|---|---|---|
| ptolA$_{strep}$ | tolA N-term streptag-II, pCDFDuet, Sm$^r$ | this study |
| ptolA$_{strep-TEV}$ | ptolA$_{strep}$ TEV site, pCDFDuet, Sm$^r$ | this study |
| pQR | ybgC-tolQ-tolR operon, pT7-1QR, Amp$^r$ | Germon et al., 1998[20] |
| ptonB$_{strepII}$ | tonB C-term TEV site and streptag-II, pACYCDuet, Cm$^r$ | this study |
| pexbB | exbB, pET26b, Kan$^r$ | Celia et al., 2016[10] |
| pexbD$_{10his}$ | exbD, C-term TEV site and 10his-tag, pCDFDuet, Sm$^r$ | Celia et al., 2016[10] |
| **strain** | | |
| W3110 | F'λ⁻IN(rrnD-rrnE)1 rph-1 | Bachmann, 1972[41] |
| ΔtolA | W3110 ΔtolA::frt | Petiti et al., 2019[42] |

followed by a streptag-II. The sequence of all plasmid constructs and mutations were verified by sequence analysis (Macrogen USA). Primer sequences for all cloning and mutagenesis experiments are available upon request.

### Protein expression
BL21(DE3) competent cells (New England Biolabs) were cotransformed with either ptolA and pQR, or ptonB, pexbB and pexbD plasmids. A single colony was grown in SEC medium with selected antibiotics, mixed with glycerol at a final concentration of 20% and stored at −70 °C. For protein expression, an aliquot of glycerol stock was spread on LB agar supplemented with antibiotics and left overnight at 37 °C. Colonies were used for a 60 ml LB starter culture, later used to inoculate 4 flasks containing 1 L 2-YT medium and incubated at 37 °C with shaking at 220 rpm until the optical density (OD$_{600}$) reached 0.8-1.0. Isopropyl β-D-1-thiogalactopyranoside (IPTG) was added at 0.1 mM final concentration and the culture was allowed to continue to grow overnight at 26 °C and 180 rpm. Cells were collected by centrifugation, resuspended in 1x phosphate buffer saline (PBS) and stored at −70 °C.

### TolA$_{TEV}$ in vivo activities
Wild type *E. coli* W3110 and ΔtolA strains were used to test the in vivo activities of tolA$_{TEV}$[41,42]. Protein production was assessed by western blot with overnight cultures. Samples (OD$_{600}$ = 0.2) were run on SDS-Page and analyzed by immunodetection with antibodies against TolA. Colicin A (Tol dependent) and D (Ton dependent) lethal activities were checked by the presence of halos on a cell lawn of the strain to be tested[43]. Sensitivity to SDS was determined by measuring bacterial growth after 4 h of incubation at 37 °C in liquid LB medium containing SDS concentration ranging from 0.25% to 2% (w/v). For division assays, cells were grown 4 h in LB liquid medium without NaCl, then immobilized on Poly-L-Lysine (0.1% w/v) coated microscope slides[42]. Observation was carried out on an Eclipse 50i optical microscope (Nikon, France).

### TolA$_{TEV}$QR purification
Cells in PBS were thawed and resuspended in 100 ml 1xPBS supplemented with 100 μM 4-(2-aminoethyl)benzenesulfonyl fluoride (AEBSF Goldbio), 100 μM DNase (Goldbio), and 50 μg/ml lysozyme (Sigma), and disrupted with two passages through an EmulsiFlex-C3 (Avestin) operating at 15,000 psi. Membranes were pelleted by ultracentrifugation in a Type 45 Ti Beckman rotor at 200,000 g for 30 min at 4 °C. Membranes were resuspended in 1 × PBS using a dounce homogenizer and solubilized by the addition of LMNG to a final concentration of 1% by stirring at medium speed for 1-2hrs at room temperature. Insoluble material was pelleted by ultracentrifugation in a Type 70Ti Beckman rotor at 300,000 g for 1 hr at 4 °C and the supernatant was used immediately.

Streptactin affinity chromatography was performed using a StrepTrap-HP 5 ml prepacked column (Cytiva) equilibrated in Tris 50 mM pH8 NaCl 200 mM EDTA 1 mM supplemented with LMNG 0.005%. Elution was performed with the same buffer supplemented with 4mM *d*-Desthiobiotin (Sigma). Fractions were analyzed by SDS-Page and those containing the bands for TolA, TolQ and TolR were pooled. TEV digestion was performed through the addition of TEV protease and DTT at 0.1 mg/ml and 2 mM final concentrations and rocked overnight at 4 °C. The soluble periplasmic domain of TolA and undigested TolAQR were eventually separated from digested TolAQR using cation exchange chromatography: the digest was diluted 10x with Tris 50 mM pH7.4 LMNG 0.005% and loaded on a prepacked Resource-S 6 ml column (Cytiva) and eluted with a linear 0-1 M NaCl gradient over 10 column volumes. The flowthrough was collected and concentrated by ultrafiltration on an Amicon-Ultra filter unit (Millipore) with a 100 kDa MW cut off. The concentrated sample was subject to SEC using a Superose6 increase 10/300 GL (Cytiva) equilibrated with Hepes/NaOH 30 mM pH7.5 NaCl 150 mM EDTA 1 mM LMNG 0.002%, yielding two elution peaks. The fractions corresponding to the second elution peak were pooled, concentrated by ultrafiltration and used for cryo-EM.

## TolAQR purification

The purification of native TolAQR, i.e. without the TEV site on TolA, is the same as for TolA$_{TEV}$QR except that the complex was solubilized from membranes with 1% DMNG. The Streptactin affinity and SEC purification steps were performed with buffers containing DMNG at 0.03% and 0.01% respectively. No cation exchange chromatography was performed.

## TonB-ExbBD purification

Cells were disrupted and membranes pelleted as described for TolA$_{TEV}$QR. The membrane pellet was resuspended in 100 ml 1x PBS and solubilized with the addition of 1% DDM by stirring at medium speed for 1-2 hrs at room temperature. Insoluble material was pelleted by ultracentrifugation in a Type 70Ti Beckman rotor at 300,000 $g$ for 1 hr at 4 °C and the supernatant was used for immobilized metal affinity chromatography. Imidazole was added to the supernatant at 20 mM final concentration and loaded on a 10 ml HisPur nickel (Thermo Fisher) prepacked column equilibrated with 1xPBS imidazole 20 mM DDM 0.1%. The column was washed in two steps using the same buffer with 40 and 60 mM imidazole and eluted with 300 mM imidazole. Fractions were analyzed by SDS-Page and those containing the three bands for TonB, ExbB and ExbD were pooled. The concentration of TonB-ExbBD in mg per ml was roughly estimated with absorbance at 280 nm reading, and a 2-3 weight per weight excess of amphipol PMAL-C12 was added to the sample and gently rocked for 30 min at room temperature. Detergent was depleted from the sample by adding freshly hydrated Bio-Beads SM-2 (Bio-Rad) and gentle stir at room temperature for 1 hr. The sample was separated from beads using a 0.22 μm vacuum-driven filter unit (Steriflip Millipore) and aggregates were pelleted by ultracentrifugation in a Type 70Ti Beckman rotor at 300,000 $g$ for 1 hr at 4 °C. The supernatant was subjected to Streptactin affinity chromatography using a StrepTrap-HP 5 ml prepacked column (Cytiva) equilibrated in Tris 50 mM pH8 NaCl 200 mM EDTA 1 mM. Elution was performed with buffer supplemented with 4mM *d*-Desthiobiotin (Sigma). SDS-Page analysis showed the presence of the TonB-ExbBD complex in the elution peak only, while the ExbBD subcomplex was found in the flowthrough. The elution and flowthrough were pooled separately and concentrated by ultrafiltration using a 100 kDa MW cut off. The concentrated TonB-ExbBD and ExbBD samples were subjected to SEC using a Superose6 increase 10/300 GL (Cytiva) equilibrated with Tris 30 mM pH8.0 NaCl 170 mM EDTA 1 mM.

## Mass Photometry

Mass photometry measurements were performed on glass cover slips using a OneMP instrument (Refeyn Ltd)[44]. The samples were diluted to 20−50 mM in 1xPBS just prior the measurement and 10 μl were deposited on the cover slip. 60 s videos were recorded using AcquireMP and analyzed with DiscoverMP (Refeyn Ltd version 2021 R1). Masses were calculated from measurements on calibrated standards.

## Sedimentation Velocity Analytical Ultracentrifugation

Sedimentation velocity was carried out at 50,000 rpm (195,650 x g at 7.0 cm) and 20 °C on a Beckman Coulter ProteomeLab XL-I analytical ultracentrifuge and An50-Ti rotor. Samples of ExbBD and TonB-ExbBD with PMAL-C12 in SEC buffer were studied in 12 mm two-channel centerpiece cells, and data were analyzed in SEDFIT[45] in terms of a continuous c($s$) distribution of sedimenting species. The solution density, viscosity, protein extinction coefficient, and protein partial specific volume were calculated in SEDNTERP. The protein refractive index increment was calculated in SEDFIT. The partial specific volume for PMAL-C12 was calculated based on its chemical composition following the method of Durchschlag and Zipper[46], and a refractive index increment of 0.14 cm³g⁻¹ was assumed. Absorbance and interference c($s$) distributions were analyzed simultaneously using the fitted $f/f_o$ membrane protein calculation module in GUSSI[47] to obtain the protein and amphipol contributions to the sedimenting complex of interest.

## EM sample preparation

3-4 microliters of TEV digested TolA$_{TEV}$QR (abs$_{280}$ = 2.3), TolAQR (abs$_{280}$ = 5) or TonB-ExbBD (abs$_{280}$ = 1.4) were applied to a Quantifoil R1.2/1.3 300 mesh grid (Electron Microscopy Sciences, Protochips, Inc.) that had been glow discharged for 45 s at 15 mA (Pelco easiGlow, Ted Pella, Inc.). The grid was blotted and immediately plunged into liquid ethane cooled down with liquid nitrogen using a Vitrobot Mark IV system (Thermo Fisher Scientific). The freezing conditions were as follows: 100% humidity, temperature 5 °C, no wait time, 2-4 secs blot time and +3 blot force.

## EM data acquisition

Data collection for native TolAQR was performed with a FEI Glacios (Thermo-Fisher) operating at 200 keV coupled with a K3 direct electron detector (Gatan) using SerialEM[48]. Micrographs were collected as dose-fractionated movies with a calibrated pixel size of 0.445 Å/pixel in the counting and super resolution mode, with 30 frames per movie and a total dose of 70 e/Å² per movie. 5563 movies were collected.

The datasets for TolA$_{TEV}$QR and TonB-ExbBD were collected on a Titan Krios G3 (Thermo-Fisher) operating at 300 keV and equipped with an Imaging Filter Quantum LS and a K3 direct electron detector (Gatan). The pixel size was 0.415 Å/pixel in super resolution mode for both datasets. For TolA$_{TEV}$QR 7,958 micrographs were collected as dose-fractionated movies with SerialEM[48] in the counting mode, with 30 frames and a total dose of 61 e/Å². For TonB-ExbBD 5784 micrographs were collected, with 46 frames and a total dose of 70 e/Å².

## EM data processing

For TolAQR the movies were imported and processed with CryoSPARC[49] v4 (Suppl. Fig. 5). The movies were binned by a factor of two (0.89 Å/px) and gain and motion corrected using patch-motion. The Ctf parameters were estimated with patch-CTF. 479 movies were rejected because of excess drift. Blob-picker was used to extract 317,684 particles from 1000 movies and submitted to 2D classification. Four 2D classes were then selected for template-picking on the 5084 movies, yielding 1,519,049 particles. The particle images were extracted with a binning of 2 (box size 128×128, 1.78 Å/px) and submitted to two rounds of 2D classification. The 217,752 selected particles were submitted to ab-initio reconstruction with three classes followed by

hetero-refinement with 3 classes. The best class comprised 126,682 particles and was subjected to non-uniform refinement, yielding a 5.9 Å resolution 3D density map. These particles were used to train a neural network for particle picking using Topaz[50]. 2,213,448 particles (box size 128×128, 1.78 Å/px) were extracted, subjected to two rounds of 2D classification. The resulting 304,787 particles were subjected to ab-initio reconstruction with five classes. The best class contained 107,631 particles and non-uniform refinement yielded a final 3D density map at 5.4 Å resolution.

For TolA$_{TEV}$QR 7958 movies were imported in Relion4[22]. The movies were gain corrected, aligned and binned by a factor of two (0.83 Å/px) using MotionCor2[51] and the CTF parameters were estimated with Gctf [52]. 7,811 exposures that showed resolution better than 4 Å were selected for further processing. A total of 11,112,982 particles were extracted with a binning of four (box size 64×64, 3.32 Å/px) using a Topaz pre-trained model. 1,694,927 particles were selected after two rounds of 2D classification. The resulting particles were re-extracted with a binning factor of two (box size 128 × 128, 1.66 Å/px) and submitted for 3D initial model with three classes. The 1,274,028 particles for the two best initial model 3D classes were selected for further processing. The best 3D initial model structure was hand flipped, aligned along a pseudo 5-fold symmetry and used as 3D reference for 3D classification into three classes. Two resulting 3D classes were selected for 3D refinement, yielding a 3.8 Å resolution density map with 811,535 particles. These particles were re-extracted (box size 256×256, 0.83 Å/px) and submitted to 3D classification with four classes and local angular searches of 20° (see Suppl. Fig. 2 for details). Three 3D classes out of four were selected, representing 474,068 particles, and 3D refinement yielded a 3.5 Å resolution map. The selected particles were submitted to Bayesian polishing, CTF refinement, followed by an extra Bayesian polishing step, resulting in a 3.27 Å resolution refined 3D map. An additional 3D classification step using a tight mask excluding the detergent micelle was performed. Of the three classes, two were selected, accounting for 59,369 and 78,674 particles respectively. These two classes were refined, then the respective particles and 3D refined maps were transferred to cryoSPARC v4. Non-uniform refinement yielded a 2.94 Å final resolution 3D density map for the 59,369 particles, and 3.18 Å for the 78,674 particles.

For TonB-ExbBD 5784 movies were imported in Relion4[22] (Suppl. Fig. 8). The movies were gain corrected and the frames were aligned with MotionCor2[51]. Defocus determination was carried out with Gctf [52]. A total of 5,414,663 particles binned with a factor of two (box size 128×128, 1.66 Å/px) were extracted using a Topaz pre-trained model. 2,788,259 particles were selected after two rounds of 2D classification and initial models were calculated using five classes. The largest initial model class (53% of the particles) contained damaged particles showing featureless structures in the membrane region. The best initial model class (26% of the particles) showed structural features expected for the ExbBD subcomplex[9] and was selected as 3D reference after correction for handedness. The 3D reference was aligned along 5-fold symmetry (C5), and C5 symmetrized. A first 3D classification step was performed using C5 symmetry to select the particles that better match the ExbB pentamer that represents the bulk of the complex. The particles were separated into four subsets, and the particles for the best class (871,353 particles) were selected for 3D refinement with C5 symmetry. The 871,353 particles were re-extracted without binning (box size 256×256, 0.83 Å/px) and used to calculate an initial 3D model with no symmetry imposed. The particles were then subjected to 3D classification using three classes. The best class accounted for 26% of the particles (227,584) that was used for 3D refinement, yielding a 3.3 Å resolution structure. The 227,584 particles were subjected to one round of Bayesian polishing followed with CTF refinement, resulting in a 3.1 Å resolution map after 3D refinement. The refined polished particles and 3D refined map were transferred to cryoSPARC v4[49] for non-uniform

refinement, yielding a 2.9 Å resolution map. A first atomic model of TonB-ExBBD was generated with ModelAngelo[53]. This model was used to generate a protein mask in Chimera[54], and a mask for the PMAL-C12 micelle using the InvertMask option. The micelle mask was smoothed and used for particle substraction in cryoSPARC. The substracted particles were submitted to local refinement using the protein mask, yielding a final 2.8 Å resolution map.

The procedure revealing TonB binding at different ExbB chains is summarized in Suppl. Fig. 9. The polished 227,584 particles used to calculate the refined 3.3 Å resolution map in relion4 were further subjected to 3D classification with local angular searches limited to 20° and a loose mask covering both the protein and the micelle. The particles were sorted into five classes, using an angular sampling of 3.7° and $T = 4$ for the first 25 iterations, then a sampling of 1.8° and $T = 8$ for an additional 20 additional iterations. After visual inspection the five resulting classes were regrouped into 3 groups (69, 18 and 13% of the particles) that showed different orientations of the ExbD dimer. Each group was 3D refined, then transferred to cryoSPARC v4, submitted to non-uniform refinement, particle substraction with a micelle mask and locally refined, yielding three structures at 3.0 Å, 3.2 Å and 3.2 Å respectively.

Except for the lower resolution, the 3.0 Å structure was the same as the 2.8 Å described previously. The two others at 3.2 Å showed the TM of TonB bound to different ExbB chains, respectively chain E or A.

### Model building and refinement

Three different structures were built: the 2.8 Å resolution structure with TonB bound to ExbB chain C, and the two 3.2 Å structures with TonB bound to either chain E or A. For ExbBD the PDB coordinates of the ExbBD Cryo-EM structure 6TYI[9] were fit into the map with UCSF ChimeraX[55] and real-space refined with Phenix[56]. For TolA$_{TEV}$QR a Model Angelo[53] partial structure generated from a sharpened map was used as a starting model. The structures were iteratively rebuilt in Coot[57] and real-space refined in Phenix. DeepEMHancer[58] maps were used in Coot for rebuilding poorer density regions. The final structures were further relaxed in ISOLDE[59].

### Reporting summary

Further information on research design is available in the Nature Portfolio Reporting Summary linked to this article.

## Data availability

The atomic coordinates of *Ec* TonB-ExbD are deposited at the Protein Data Bank under the accession codes 9DDO (TonB bound to ExbB chain C), 9DDP (TonB bound to ExbB chain E) and 9DDQ (TonB bound to ExbB chain A). For the *Ec* TolA$_{TEV}$QR structures the accession codes are 9DDM (TolA$_{TEV}$QR-I) and 9DDN (TolA$_{TEV}$QR-II). The associated cryoEM 3D maps are deposited at the Electron Microscopy Data Bank under accession codes EMD-46778 (TonB bound to ExbB chain C), EMD-46779 (TonB bound to ExbB chain E), EMD-46780 (TonB bound to ExbB chain A), EMD-46776 (TolA$_{TEV}$QR-I), EMD-46777 (TolA$_{TEV}$QR-II) and EMD-70142 (TolAQR without TEV sequence). Data supporting the findings of this study are available upon request. The biochemical and biophysical data generated in this study are provided in the Supplementary information. The source data underlying Supplementary Figs. 1, 6 and 7 are provided as a Source Data File. Source data are provided with this paper.

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

## Acknowledgements

The authors thank Yanxiang Cui, Huaibin Wang and Ulrich Baxa for technical support on the NIH MICEF electron microscopes, and Di Wu and Grzegorz Piszczek for support with biophysical analyses. H.C., I.B., R.G., B.M.B. and S.K.B. are supported by the Intramural Research Program of the National Institute of Diabetes and Digestive and Kidney Diseases, NIH. D.D. and R.L. are supported by the Centre National de la Recherche Scientifique, the Aix-Marseille Université and grants from the Agence Nationale de la Recherche (ANR-18-CE11-0027). This work utilized the NIH Multi-Institute Cryo-EM Facility (MICEF), the computational resources of the NIH HPC Biowulf cluster (http://hpc.nih.gov) and the Biophysics Core Facility (NHLBI).

## Author contributions

H.C., R.L. and S.K.B. conceived the study. H.C. designed mutants, expressed and purified the proteins, prepared EM grids, collected and analyzed cryo-EM data, and wrote the manuscript. I.B. constructed the structures and wrote the manuscript. R.G. collected and analyzed SV-AUC data. D.D. tested the in vivo activities of the tolA$_{TEV}$ construct. B.M.B and R.L designed expression plasmids. S.K.B. directed the work and wrote the manuscript.

## Funding

## Competing interests

The authors declare no competing interests.
