## [Transparent Peer Review file · Nature Communications]

Cryo-EM structures of the E. coli Ton and Tol motor complexes

Corresponding Author: Dr Susan Buchanan

Version 0:

Reviewer comments:

Reviewer #1

(Remarks to the Author)

This work focuses on the Ton and Tol protein complexes that are two bacterial molecular motors coupling the proton gradient across the inner membrane to generate energy. This energy is transmitted through the periplasmic space to outer membrane-associated proteins, supporting different functions: the Tol system stabilizes outer membrane integrity, while the Ton system is responsible for active import of nutrient.

The structural data on these two molecular motors remain incomplete, and their mechanisms of energy production and transfer are not yet understood.

Here the authors used cryoEM to study the structures of the E. coli TonB-ExbB-ExbD and TolA-TolQ-TolR complexes. They provide a detailed description of the inner membrane parts of both machineries, while the periplasmic globular domains, and the disordered domains connecting them to the membrane remain unresolved. The authors describe structural similarities between the two complexes, suggesting a shared mechanism of action. Additionally, they observe that TonB and TolA interact selectively with specific subunits of ExbB and TolQ respectively, and propose a sequential assembly model that redefines how these complexes are organized and function.

The structural data are thoroughly analyzed, and the proposed model for rotation and proton channel opening and closing are interesting. My major concern is the absence of any mutation and phenotypic assays to support the proposed models. These additional experiments would provide stronger support for the mechanisms suggested by the structural data.

Another major concern is the absence of the periplasmic domains in these structures, as these regions have also been undetected in other structures of homologous Ton/ Tol complexes. In this context, what was the rationale for cleaving TolA at residue 50 which marks the beginning of the conserved R-binding motif? This motif is predicted to interact with TolR and is highly conserved in TolA/TonB protein family, as also noted by the authors, line 401.

Did the author attempt other constructs including the R motif?

Minor points:

-Have lipids been observed in other TolQR/ExbBD structures? If so, it would be valuable to discuss and compare the types of lipids and their binding sites.

-Please specify either in Materials and methods or in line 102 that TolATEV comprises the residues 1-50 plus X residues (which ones?) corresponding to the TEV cleavage site.

-Supplementary Figure 3: Pro 190 label appears cropped.

-Line 208 should reference Figure 3, not Figure 2.

-In Supplementary Figure 4, why Proline is red?

-In the legend of Figure 4, please provide the color code for panel C.

Reviewer #2

(Remarks to the Author)

The manuscript by Celia et al. reports the cryo-EM structures of E.coli TonB-ExbB-ExbD and TolA-TolQ-TolR complexes. The higher resolution obtained, compared to the previous one, allowed to build membranes-embedded region of TolA and TonB and reveal how they interact with TolQ and ExbB, respectively.

The authors provide a detailed analysis of the structures and describe differences between the subunits within the complexes. One of the strengths of the manuscript is the analysis of different data sets and identification of different population through further subclass analysis, which reveals the dynamics of the complex and the preferential binding of TonB to some ExbB subunits. Authors postulate that upon interaction with the outer-membrane receptor, interactions in the periplasmic region between TonB's D-box and ExbD are responsible for the subsequent preferential binding of TonB to some of ExbB subunits.

In the discussion, authors use the comparison of the two systems combined with an analysis of conserved residues and features to suggest a common mechanism between TonB-ExbB-ExbD and TolA-TolQ-TolR complexes. They further suggest that conformation changes in the TonB TM domain is transmitted to ExbB subunits via key conserved residues and could trigger the opening of the channel necessary to allow the rotation of ExbD or TolR and the transfer of protons. However, the later stages of this mechanism remain unclear.

The manuscript is clearly written, and the results are well presented, providing additional structural information that enhances the understanding of the system. For all these reasons, I believe this article is suitable for publication in this journal

Minor comments:

-L 103 and Supplementary figure 1. Information about the first peak of TolATEVQR and the shoulder of TolAQR on the SEC profiles are not provided.

-L. 157 Residues 4 to 34 'of TolA'

-L.210 photometry instead spectrometry

- Some figures are hard to interpret:

Figure 2D: The link between the two panels is not indicated. Is it a 180° rotation? A different orientation or rainbow coloring might help to distinguish the helices.

Figure 2F is unclear, Figure 2G is probably sufficient

Figure 4D: This panel is not clear. what are the yellow circles?

Supplementary figure 3: The colors in the figure do not seem to match the description. This maybe a pdf conversion issue?

Reviewer #3

(Remarks to the Author)

In this work, Celia and co-workers report the structure of bacterial Ton and Tol motor complexes. The cryoEM structures determined to 2.8 and 3 angstrom resolution provide insight into the structure and assembly of these motor complexes. I have a few comments for the authors.

The introduction would benefit from a more detailed introduction to the two motor complexes, which is particularly important for the broad readership of Nature Communications. For example, what is known/reported about stoichiometry. Along these lines, some of the discussion (which reads like an introduction in some parts) can be moved, such as the first paragraph.

Essential to obtaining the TolAQR complex was the introduction of a TEV protease site in TolA. The location of this cut site can be added to Figure 1. Moreover, after TEV digestion the complex has substantially reduced when comparing 2D classifications in fig S1. Does the TEV cut complex still support function? I understand this helped structural analysis, but further justification is needed.

Only two TolA subunits are observed in the TEV digested TolAQR complex. However, I find this perplexing as what is preventing a 1:1 stoichiometry of TolA:TolQ. Is this an artefact of the TEV protease cleavage? A low-resolution reconstruction of the TolAQR complex is presented in Fig S5. No density for the periplasmic region is shown yet the 2D class averages show distinct features here. Why? In addition, a flow chart for processing of the LMNG data is not provided.

The authors have used mass "photometry" not "spectrometry" as noted in results section (line 210). The mass distribution is quite broad as expected for MP. The authors should update fig s6 with lines to indicate specific stoichiometries. Based on this mass range, other stoichiometries would be difficult to rule out. Same for the AUC data, denote expected values for specific stoichiometries.

Following on the point above, similar data for TolAQR (+/- TEV digestion) is not provided. The authors should add this data to the paper. The presented gel filtration traces are broad suggestive of polydisperse stoichiometries.

While I appreciate the structural work, the proposed model (fig 4e) of Ton assembly and mechanism is not warranted. Figure 1 presents TonB-ExbDB followed by the TolAQR complex. However, these complexes are presented in the reverse order in the paper.

Version 1:

Reviewer comments:

Reviewer #1

(Remarks to the Author)

In the revised version, the authors have improved the overall quality of the text and figures.

Regarding my major concern about the validation of their mechanistic model through mutation and phenotypic assays, they have provided a list of mutants from the literature (designed and tested by other teams) along with their corresponding phenotypes. This information supports the mechanistic model presented by the authors in this manuscript.

However, I still have a major concern regarding TolAQR structure. This structure was obtained using a TolA domain that was designed without the motif of interaction with TolR, a motif that is highly conserved in the TolA/TonB protein family. The author's response, "we did not know at the time we designed this construct that this motif was important" is not convincing. This region is crucial for the binding, and its absence can modify the binding of TolA to its partner, thereby affecting the stoichiometry of the assembly. In addition, the cleaved version of TolA has 16 additional residues that are not part of the original sequence of the protein, corresponding to the Strep tag and TEV cleavage sites.

The stoichiometry of the presented structure 2TolA/5TolQ/2TolR could have been a significant and novel result of this manuscript, potentially revealing a different stoichiometry and mode of interaction compared to the Ton complex. However, it seems that this stoichiometry and assembly may be an artifact, as the authors themselves state that "The main difference between the TonB-ExbBD and TolAQR complexes is the respective stoichiometries of TonB and TolA. The physiological significance of two TolA subunits in TolAQR is not clear." And also they note "TEV digested TolATEVQR is mostly monomeric (suppl fig 1-DE), suggesting that interactions between the TolA periplasmic domains might influence the oligomeric state of TolAQR." Additionally, the SEC elution peaks of the TolAQR display heterogeneity and varying proportions of mono/ dimer species between the different forms of the complex with cleaved or full-length version of TolA. In conclusion, this manuscript presents high resolution structural data on Ton and Tol complexes. The data supports the phenotypic assays that have been published by other teams. The TolAQR structure which could have provided valuable insight, appears to be an artifact of cleavage, and more critically lacks the conserved residues involved in the interaction with TolR, which might be essential for the stability and the stoichiometry of the complex. Although the structures presented in this manuscript do not reveal any unprecedented features, their detailed analysis are of interest to specialists in the field of Ton and Tol machineries.

Reviewer #2

(Remarks to the Author)

I am happy with the responses provided to the reviewers' comments and the additional data included

Reviewer #3

(Remarks to the Author)

The authors have addressed my concerns and recommend for publication.

REVIEWER COMMENTS

Reviewer #1 (Remarks to the Author):

This work focuses on the Ton and Tol protein complexes that are two bacterial molecular motors coupling the proton gradient across the inner membrane to generate energy. This energy is transmitted through the periplasmic space to outer membrane-associated proteins, supporting different functions: the Tol system stabilizes outer membrane integrity, while the Ton system is responsible for active import of nutrient.

The structural data on these two molecular motors remain incomplete, and their mechanisms of energy production and transfer are not yet understood.

Here the authors used cryoEM to study the structures of the E. coli TonB-ExbB-ExbD and TolA-TolQ-TolR complexes. They provide a detailed description of the inner membrane parts of both machineries, while the periplasmic globular domains, and the disordered domains connecting them to the membrane remain unresolved. The authors describe structural similarities between the two complexes, suggesting a shared mechanism of action. Additionally, they observe that TonB and TolA interact selectively with specific subunits of ExbB and TolQ, respectively, and propose a sequential assembly model that redefines how these complexes are organized and function.

The structural data are thoroughly analyzed, and the proposed model for rotation and proton channel opening and closing are interesting. My major concern is the absence of any mutation and phenotypic assays to support the proposed models. These additional experiments would provide stronger support for the mechanisms suggested by the structural data.

We thank the reviewer for their comments. We agree that mutations and phenotypic assays are valuable tools. Most of the residues of the Ton and Tol complexes that are potentially involved in proton translocation, or the rotation mechanism, have been mutated and their associated phenotypes described in several reports.

In our hypothetical model for the opening of the proton channel, a re-orientation of the conserved Ser and His on the TonB/TolA TM domain is propagated through direct contact with the conserved Ser and Trp on ExbB/TolQ TM1 and then to the conserved Pro residue on TM3, to trigger the re-orientation of TM3 and the opening of the channel. Point mutations for all these key residues have been described, and most have dramatic effects on Ton or Tol activities that are dependent on the proton motive force. We have added a supplementary table that summarizes the reported point mutations and their associated phenotypes, and added a sentence in the discussion section:

(line 375) “In this configuration, a change of orientation of the TM of TonB or TolA could modify the conformation of ExbB_C Pro190 or TolQ_{C,E} Pro187 via the two conserved Ser and Trp residues on TM1. *Most of these residues are critical as their mutations result in the inhibition or a complete loss of pmf dependent Ton and Tol activities (suppl table 1).*”

Concerning the rotation, this mechanistic model has been first introduced by the Susan Lea and Nicholas Taylor labs based on the cryoEM structures of the MotAB stator complex (Deme *et al.*, Santiveri *et al.*, 2020). As the FliG rotor protein rotates together with the flagellum, it was hypothesized that the MotA pentamer rotates around the peptidoglycan anchored MotB dimer and transmits the rotation to FliG. Because of the extensive structural homologies between MotAB, ExbBD and TolQR, we have adapted the rotation model to the Ton (Ratliff, Celia and Buchanan, 2022) and Tol systems, in which the rotation of ExbD or TolR relative to ExbB or TolQ leads to the pulling of the periplasmic domain of TonB or TolA anchored to outer membrane components. The number of publications describing mutations in the hydrophobic pore region of the different complexes is too extensive to report here.

Another major concern is the absence of the periplasmic domains in these structures, as these regions have also been undetected in other structures of homologous Ton/ Tol complexes. In this context, what was the rationale for cleaving TolA at residue 50 which marks the beginning of the conserved R- binding motif? This motif is predicted to interact with TolR and is highly conserved in TolA/TonB protein family, as also noted by the authors, line 401.

Did the author attempt other constructs including the R motif?

We share the frustration of reviewer 1 that the periplasmic domains are absent in our structures. As we analyzed the cryoEM data we have explored different approaches using several masks and particle subtraction followed with 2D and/or 3D classification, but the periplasmic domains remain invisible, likely due to their high flexibility compared to the membrane embedded part of the complex. As for the rationale for cleaving TolA at residue 50, we did not know at the time we designed this construct that this motif was important. In future work we plan to keep the R-motif intact.

To determine whether the introduction of the TEV site was detrimental, we have now tested *in vivo* activities of the tolA_{TEV} construct and show that tolA_{TEV} is as active as wt tolA. The results are summarized in a new supplemental figure (suppl fig 6), they include sensitivity to the Tol dependent colicin A, growth in the presence of SDS, and impact on cell division. Denis DUCHE has performed the *in vivo* test activities and been added as a coauthor. The material and methods section has been updated, and the result section *Stoichiometry of the Ec TolAQR complex* has been modified:

(line 183) “The stoichiometry of the TolA_{TEV}QR complex is two TolA, five TolQ and two TolR. The structure was obtained with a truncated version of TolA containing a TEV sequence inserted between residues Ile50 and Asp51, which lies in the conserved motif Ile50-Asp-Ala-Val-Met-Val-Asp56 (IDAVMVD) that is predicted to interact with TolR⁷. This modification could affect the assembly **and/or activity** of the TolAQR complex.

A cryo-EM dataset of native, full-length TolAQR in DMNG was collected at 200keV and SPA yielded a 5.4Å resolution structure calculated with 107,631 particles (suppl fig 5-A). A perfect fit was found between this map and the TolA_{TEV}QR structure, showing that insertion of the TEV site on TolA did not affect the stoichiometry of the complex (suppl fig 5-B).

Furthermore, the insertion of the TEV sequence had no significant effect on Tol activities. Using a Δ tolA strain, we found that complementation with the tolA_{TEV} construct or wt tolA restored all Tol dependent activities, i.e. ability to grow on SDS containing medium, sensitivity to the Tol dependent colicin A, and absence of filamentous cells during cell division (suppl fig 6).”

Minor points:

-Have lipids been observed in other TolQR/ExbBD structures? If so, it would be valuable to discuss and compare the types of lipids and their binding sites.

Lipids have not been observed in TolQR structures. We were able to model three PE (phosphatidyl ethanolamine) and one PG (phosphatidyl glycerol) molecules in our previously published ExbBD structure in nanodisc (pdb 6TYI). For the TonB-ExbBD complex, two PE molecules were modelled. Mass spectrometry has been performed on purified *Sm* ExbBD, showing that PG, PE and CL (cardiolipin) copurified with the complex, and five PG molecules were modelled for the five-fold symmetrized map of the ExbB pentamer (Biou *et al.*, 2022). All these tightly bound lipids are in the inner leaflet of the cytoplasmic membrane and seem to be anchored through interaction with Arg200 for *Ec* ExbB, and Arg237 for *Sm* ExbB. We have added a sentence in the results section describing the bound lipids:

(line 261) “No external lipids were added during purification. *As observed for Sm ExbB (pdb 6YE4¹⁶), these tightly bound lipids are anchored through interaction between their polar head group and an arginine residue (Arg200 for Ec ExbB, Arg237 for Sm ExbB).*”

-Please specify either in Materials and methods or in line 102 that TolATEV comprises the residues 1-50 plus X residues (which ones?) corresponding to the TEV cleavage site.

We added this information in the “Bacterial strains and plasmids” method section:

(line 479) “The mutant of *EctolA* with a TEV site insertion was prepared by site-directed mutagenesis using the Q5 Site-Directed Mutagenesis Kit (New England Biolabs). *Following TEV digestion, the sequence of the resulting N-terminal fragment is*

MGSWSHPQFEKGSSKATEQNDKLRRAIIISAVLHVILFAALIWSSFDENIEASAGGGGGSSIE
NLYFQ (streptag-II and TEV sequences are underlined).”

-Supplementary Figure 3: Pro 190 label appears cropped.

Fixed.

-Line 208 should reference Figure 3, not Figure 2.

Fixed.

-In Supplementary Figure 4, why Proline is red?

Thank you for pointing that out. Pdbsum uses orange for proline. We have modified the color of Pro12 in the figure, and updated the color code in the figure legend.

-In the legend of Figure 4, please provide the color code for panel C.

Done, for panel D as well.

Reviewer #2 (Remarks to the Author):

The manuscript by Celia et al. reports the cryo-EM structures of E.coli TonB-ExbB-ExbD and TolA-TolQ-TolR complexes. The higher resolution obtained, compared to the previous one, allowed to build membranes-embedded region of TolA and TonB and reveal how they interact with TolQ and ExbB, respectively.

The authors provide a detailed analysis of the structures and describe differences between the subunits within the complexes. One of the strengths of the manuscript is the analysis of different data sets and identification of different population through further subclass analysis, which reveals the dynamics of the complex and the preferential binding of TonB to some ExbB subunits. Authors postulate that upon interaction with the outer-membrane receptor, interactions in the periplasmic region between TonB's D-box and ExbD are responsible for the subsequent preferential binding of TonB to some of ExbB subunits. In the discussion, authors use the comparison of the two systems combined with an analysis of conserved residues and features to suggest a common mechanism between TonB-ExbB-ExbD and TolA-TolQ-TolR complexes. They further suggest that conformation changes in the TonB TM domain is transmitted to ExbB subunits via key conserved residues and could trigger the opening of the channel necessary to allow the rotation of ExbD or TolR and the transfer of protons. However, the later stages of this mechanism remain unclear.

The manuscript is clearly written, and the results are well presented, providing additional structural information that enhances the understanding of the system. For all these reasons, I believe this article is suitable for publication in this journal

We thank the reviewer for their positive comments.

Minor comments:

-L 103 and Supplementary figure 1. Information about the first peak of TolATEVQR and the shoulder of TolAQR on the SEC profiles are not provided.

CryoEM analysis of LMNG solubilized TolAQR complexes shows a large heterogeneity of oligomeric states, reflecting the broadening of the elution “peak” (suppl fig S1-C). We do not know the nature of the shoulder on the SEC profile. The first peak of TolA_{TEV}QR contains a mixture of monomers and dimers of the complex. CryoEM analysis of dimers had poor resolution, therefore we focused on the second peak that showed mostly monomers. The results section has been updated to explain the first peak of TolA_{TEV}QR:

(line 116) “This new construct copurified with TolQ and TolR and SEC yielded two elution peaks after TEV proteolysis (suppl fig 1-AB). **The first SEC elution peak contained a mixture of dimers and monomers of the complex, but analysis of the dimer particles only yielded low resolution 3D structures (data not shown).** Cryo-EM images showed homogeneous particles **of monomers** for the second SEC elution peak (suppl fig 1-D), which was used for **high resolution** cryoEM data collection and image analysis.”

-L. 157 Residues 4 to 34 ‘of TolA’

Fixed.

-L.210 photometry instead spectrometry

Fixed.

- Some figures are hard to interpret:

Figure 2D: The link between the two panels is not indicated. Is it a 180° rotation? A different orientation or rainbow coloring might help to distinguish the helices.

The link between the two panels is 90° and is now indicated on the figure. The helices are now colored using rainbow.

Figure 2F is unclear, Figure 2G is probably sufficient

Figure 2F has been removed. The figure legend has been updated accordingly.

Figure 4D: This panel is not clear. what are the yellow circles?

The yellow circles represent the TM domains of TonB. The figure legend has been updated to better describe the figure:

(line 1024) “D. Two schematic representations of the TonB-ExbBD complex, seen from the periplasm. ExbB subunits shown as rounded triangles, TonB (yellow) and ExbD (black and light grey) TMs as circles, ExbD periplasmic domains as ovals, the TonB linker connecting the TM and the D-box with yellow dashed lines, and the D-box as a yellow arrow. The distal C-terminal domain of TonB is not represented. The color code is the same as in C. The upper panel has the D-box bound to the ExbD periplasmic dimer in one orientation, leading to the binding of the TonB TM domain either to ExbB_E (orange) or ExbB_A (blue). In the lower panel the D-box is in the opposite direction, and the TonB TM domain binds ExbB_C (green).”

Supplementary figure 3: The colors in the figure do not seem to match the description. This maybe a pdf conversion issue?

We do not see this mismatch in our pdf document.

Reviewer #3 (Remarks to the Author):

In this work, Celia and co-workers report the structure of bacterial Ton and Tol motor complexes. The cryoEM structures determined to 2.8 and 3 angstrom resolution provide insight into the structure and assembly of these motor complexes. I have a few comments for the authors.

The introduction would benefit from a more detailed introduction to the two motor complexes, which is particularly important for the broad readership of Nature Communications. For example, what is known/reported about stoichiometry. Along these lines, some of the discussion (which reads like an introduction in some parts) can be moved, such as the first paragraph.

We thank the reviewer for their comments.

We have added a paragraph in the introduction that discusses the reported stoichiometries of the ExbB-ExbD subcomplex:

(line 59) “Numerous structures of TBDTs have been reported, along with ExbBD subcomplexes and periplasmic fragments of TonB and ExbD^{5,6}. Recent crystallographic studies revealed how TonB and ExbD interact in the periplasm^{7,8}. The stoichiometry of the ExbBD subcomplex has been a matter of debate as ExbB/ExbD ratios of 4/2, 5/1, 6/3 and 5/2 have been reported^{9,10,11,12}. A four stage model has also been proposed for the Ton activity, in which the oligomeric ratio of ExbB, ExbD and TonB is different for each stage¹³. However, the 5/2 ratio is likely to

correspond to the physiological state as it has also been found for *S. marcescens* ExbBD, *E. coli* TolQR, and the closely related MotAB complexes that use the pmf to drive the flagellar rotation^{14, 15, 16, 17, 18}.

For the Tol system, most of the structural knowledge comes from soluble fragments or complexes of periplasmic domains of Pal, TolB, TolR and TolA (see ²). As mentioned above, a 4.3Å resolution cryo-EM structure of *Ec*TolQR shows the five TolQ to two TolR stoichiometry, but the low resolution of the map did not allow structure building with high confidence¹⁸.”

The first paragraph of the discussion section has been moved to the introduction and edited:

(line 73) “The ExbBD and TolQR subcomplexes are highly homologous. TonB and TolA are less similar, reflecting their different functions, but do have in common the conserved SHLS motif (Ser-X₃-His-X₆-Leu-X₃-Ser) in their TM domain that is involved in the interaction of TonB with ExbBD and TolA with TolQR^{19, 20}. The two systems are similar to the point that they are able to complement each other, meaning that some Ton and Tol activities are still detected in *exbBD* or *tolQR* knock out strains, but absent in a double knockout¹. This cross-complementation suggests that TonB can interact with TolQR and TolA with ExbBD, and that the use of the pmf and propagation of force to TonB and TolA are similar.”

Essential to obtaining the TolAQR complex was the introduction of a TEV protease site in TolA. The location of this cut site can be added to Figure 1. Moreover, after TEV digestion the complex has substantially reduced when comparing 2D classifications in fig S1. Does the TEV cut complex still support function? I understand this helped structural analysis, but further justification is needed.

The TEV site location has been added in figure 1.

The magnification for the 2D class images is different for panels C and D, but the size of the TolAQR complexes is actually the same in the two panels. Scale bars have been added to avoid confusion.

After TEV digestion the complex is no longer active as the C-terminal domain of TolA, that is essential for activity, is no longer connected with TolQR. To verify the impact of the TEV insertion on TolA, we have conducted *in vivo* test activities and showed that the *tolA*_{TEV} construct is as active as wt *tolA*. These new results are summarized in the new supplementary figure S6.

Only two TolA subunits are observed in the TEV digested TolAQR complex. However, I find this perplexing as what is preventing a 1:1 stoichiometry of TolA:TolQ. Is this an artefact of the TEV protease cleavage? A low-resolution reconstruction of the TolAQR complex is presented in Fig S5. No density for the periplasmic region is shown yet the 2D class averages show distinct features here. Why? In addition, a flow chart for processing of the

LMNG data is not provided.

There are two TolA subunits in our structures of TEV digested TolA_{TEV}QR in LMNG and TolAQR in DMNG (as shown in suppl fig S10-D we also see a weak density for a third TolA subunit for TolA_{TEV}QR). For the TonB-ExbBD structure we only see one TonB subunit. As discussed in the manuscript we also wondered what prevented a 1:1 stoichiometry of TolA:TolQ and TonB:ExbB since “the regions of ExbB or TolQ involved in the interaction with TonB or TolA are equally exposed in the membrane and share the same conformation” (lines 398-399). This cannot be an artefact of the TEV protease cleavage since the same stoichiometry is observed in the structure of the intact TolAQR in DMNG and TolA_{TEV}QR in LMNG.

We do not observe 3D densities for the periplasmic regions in either of the TolAQR, TolA_{TEV}QR and TonB-ExbBD cryoEM structures, suggesting these regions are highly flexible. We are not sure which average 2D class the reviewer refers to. Our analysis is that none of the 2D class averages have visible density for the periplasmic regions. Only one average 2D class shows extra density on both sides of the micelle (2D class average in first row, third column of fig S1-C). We actually believe that this 2D class corresponds to a trimer or tetramer of TolAQR complexes that show opposite orientations in the micelle. In this case the extra density corresponds to the TolQ cytoplasmic domains.

We have added a flow chart for the processing of the DMNG data as figure S5-A (LMNG was indicated by mistake, DMNG was used to solubilize the complex).

The authors have used mass “photometry” not “spectrometry” as noted in results section (line 210). The mass distribution is quite broad as expected for MP. The authors should update fig s6 with lines to indicate specific stoichiometries. Based on this mass range, other stoichiometries would be difficult to rule out. Same for the AUC data, denote expected values for specific stoichiometries.

We have corrected the word “photometry”. We have added the estimated molecular weights on fig S7-C (previously S6).

Our mass photometry experiments show a major single peak for both ExbBD and TonB-ExbBD, suggesting we do not have a mixed population with different stoichiometries for each complex. Mass photometry provides us with an estimated mass of the complex, corresponding to the protein together with the associated amphipol PMAL-C12. Since we do not know the contribution of the amphipol to the mass it is difficult to estimate the position of the peaks for different stoichiometries. Assuming that the ExbBD subcomplex has a ExbB:ExbD 5:2 ratio, the 34kDa mass difference between the two peaks suggests there is only one TonB in the TonB-ExbBD complex (MW of TonB_{strep} is 28.5kDa).

As for the AUC data, it is challenging to predict the sedimentation coefficient value as it depends not only on the stoichiometry of the complex, but also the amount of detergent or amphipol associated, its shape etc. The presence of a main single peak for both complexes suggests we have a homogenous population and the estimates contributions of the different subunits and amphipols are in good agreement with the expected

stoichiometries of the complexes. We have added in the legend the protein and amphipol mass contributions as calculated from the sedimentation data for the main peak:

(100) “D. Sedimentation velocity absorbance (top panels) and interference (bottom panels) profiles for TonB-ExbBD (left panels) and ExbBD (right panels) in PMAL-C12 showing a major species at 7.1 S corresponding to the expected complexes. For TonB-ExbBD the calculated protein mass is 196 ± 26 kDa (expected mass for a 1:5:2 complex is 195.651 kDa) and the mass of the complex (protein + PMAL-C12) is 214 ± 30 kDa. For ExbBD, the calculated protein mass is 169 ± 24 kDa (the expected mass for a 5:2 complex is 167.340 kDa) and the mass of the complex (protein + PMAL-C12) is 186 ± 28 kDa.”

Following on the point above, similar data for TolAQR (+/- TEV digestion) is not provided. The authors should add this data to the paper. The presented gel filtration traces are broad suggestive of polydisperse stoichiometries.

As pointed out in figure S1, the broadening of the peak for the TolAQR gel filtration trace is likely the result of different oligomerization states of the complex, which is illustrated by the cryoEM images in S1-C. We have performed SV-AUC and MP experiments with the TolAQR complex but the collected data did not allow us to reliably determine the stoichiometry of the complex. We believe this is due to the variability of the detergent micelle size, and/or the partial occupancy of a third TolA subunit. Furthermore, we did not have the TolQR complex to compare with TolAQR.

While I appreciate the structural work, the proposed model (fig 4e) of Ton assembly and mechanism is not warranted.

Figure 1 presents TonB-ExbDB followed by the TolAQR complex. However, these complexes are presented in the reverse order in the paper.

The model shown in fig 4-E is still hypothetical. However, this is the best model we could come up with based on our knowledge of Tol and Ton. The rotation model of TolR/ExbD relative to TolQ/ExbB is derived from the rotation mechanism proposed for the homologous MotAB stators (Deme *et al.*, Santiveri *et al.*, 2020) and adapted to Ton (Ratliff, Celia and Buchanan, 2022). The assembly of TonB and TolA to ExbBD and TolQR is derived from our observation of preferential binding between TolA and TolQ and TonB and ExbB. Figure 1 now shows the complexes in the right order.

Reviewers' comments:

Reviewer #1 (Remarks to the Author):

In the revised version, the authors have improved the overall quality of the text and figures.

Regarding my major concern about the validation of their mechanistic model through mutation and phenotypic assays, they have provided a list of mutants from the literature (designed and tested by other teams) along with their corresponding phenotypes. This information supports the mechanistic model presented by the authors in this manuscript.

We thank the reviewer and are glad we addressed one of their concerns.

However, I still have a major concern regarding TolAQR structure. This structure was obtained using a TolA domain that was designed without the motif of interaction with TolR, a motif that is highly conserved in the TolA/TonB protein family. The author's response, "we did not know at the time we designed this construct that this motif was important" is not convincing.

We believe there must be a misunderstanding as we show in the original version of our manuscript that the TolAQR complex, which does not have the TEV sequence inserted, and the TEV digested TolA_{TEV}QR complex yield identical structures as shown in suppl fig 5-B. CryoEM single particle analysis of the TolAQR complex in DMNG resulted in a 5Å resolution 3D density map that shows the same stoichiometry and architecture as the 3Å resolution TolA_{TEV}QR structure. Since there is no TEV sequence on TolA, the R-box motif is intact in this complex and this map corresponds to the native structural state of the complex. As shown in suppl fig 5-B, the 5Å map perfectly matches the atomic model derived from the 3Å resolution map of the TolA_{TEV}QR complex, with clear densities for the five TolQ, the dimer of transmembrane domain of TolR in the center of the TolQ pentamer, and the two transmembrane domains for two TolA at the periphery of the pentamer. The native TolAQR and TolA_{TEV}QR complexes have thus the same 2:5:2 TolA:TolQ:TolR stoichiometry and the same 3D architecture, showing that the high-resolution structure of TolA_{TEV}QR is not artefactual. Perhaps Reviewer #1 thought that the 5Å structure was the TolA_{TEV}QR complex that was not TEV proteolyzed? To avoid any confusion, we have added in the revised version of the manuscript a sentence stating that the TolAQR structure does not have the TEV site inserted on TolA.

Lines 188-192:

A cryo-EM dataset of native, full-length TolAQR in DMNG was collected at 200keV and SPA yielded a 5.4Å resolution structure calculated with 107,631 particles (suppl fig 5-A). **This complex does not have the TEV sequence inserted in TolA and therefore has the intact R-box motif.** A perfect fit was found between this map and the TolA_{TEV}QR structure, showing that insertion of the TEV site on TolA did not affect the stoichiometry of the complex (suppl fig 5-B).

We are including with this revision of our manuscript the 5Å resolution 3D map of the TolAQR complex that does not have the TEV sequence, in case reviewer #1 would like to compare this map with the high-resolution structure. We also provide the coordinates for the TolA_{TEV}QR structure fitted into the 5Å map (TolAQR_fit_5A.pdb). The 5Å resolution map has now been deposited in the Electron Microscopy Data Base under the EMD-70142 code and this information was added to table 1 and the Data availability section.

In addition, the cleaved version of TolA has 16 additional residues that are not part of the original sequence of the protein, corresponding to the Strep tag and TEV cleavage sites.

As shown in suppl fig 6, the presence of the streptag and TEV sequences has no effect on the physiological Tol activities, suggesting that they do not interfere with a correct assembly of a functional TolAQR complex. We believe that the R-box is still functional in the construct with the TEV sequence: the TEV sequence was inserted between the Ile and Asp of the IDAVMVDS R-box motif, and the remaining DAVMVDS is probably still able to bind the TolR periplasmic dimer. The TEV proteolysis step is performed in the later stages of purification with a fully assembled solubilized complex, and we have no reason to believe that the proteolysis would affect the stoichiometry. The fact that the 5Å resolution structure of the complex without the TEV site is the same as that of the TolA_{TEV}QR after TEV proteolysis shows that the 3Å resolution structure is not an artefact due to the presence of the TEV sequence.

The stoichiometry of the presented structure 2TolA/5TolQ/2TolR could have been a significant and novel result of this manuscript, potentially revealing a different stoichiometry and mode of interaction compared to the Ton complex. However, it seems that this stoichiometry and assembly may be an artifact, as the authors themselves state that “The main difference between the TonB-ExbBD and TolAQR complexes is the respective stoichiometries of TonB and TolA. The physiological significance of two TolA subunits in TolAQR is not clear.” And also they note “TEV digested TolA_{TEV}QR is mostly monomeric (suppl fig 1-DE), suggesting that interactions between the TolA periplasmic domains might influence the oligomeric state of TolAQR.” Additionally, the SEC elution peaks of the TolAQR display heterogeneity and varying proportions of mono/ dimer species between the different forms of the complex with cleaved or full-length version of TolA.

This concern about the oligomerization of the TolAQR complex was not raised in the first comments of reviewer #1. The SEC elution peak of the TolAQR complex, either in LMNG or DMNG, shows a wide distribution, either with the native TolAQR or TolA_{TEV}QR complexes, showing that it is not a consequence of the TEV site insertion. This is in part due to the presence of the very elongated TolA periplasmic domain, but also to the presence of different oligomers of TolAQR complex as shown on suppl fig 1-C. Because extended oligomers are not detected for the TolA_{TEV}QR complex after TEV digestion, it suggests that the periplasmic domain of TolA participates in the oligomerization. This is

an observation we did not comment further as the periplasmic domain of TolA is not visible in the 3D map of the native TolAQR complex. In fact, the periplasmic domain is not visible in any of the motor complex structures published so far.

In conclusion, this manuscript presents high resolution structural data on Ton and Tol complexes. The data supports the phenotypic assays that have been published by other teams. The TolAQR structure which could have provided valuable insight, appears to be an artifact of cleavage, and more critically lacks the conserved residues involved in the interaction with TolR, which might be essential for the stability and the stoichiometry of the complex. Although the structures presented in this manuscript do not reveal any unprecedented features, their detailed analysis are of interest to specialists in the field of Ton and Tol machineries.

The data presented in the manuscript show that the stoichiometry 2TolA/5TolQ/2TolR is not an artefact due to the insertion of the TEV sequence on TolA. Therefore, we think it represents a significant and novel result. Our structures are unprecedented as we present the first structures of both the full Ton and Tol motor complexes, showing not only how TolA and TonB interact with TolQR and ExbBD respectively, but also that they share a common mode of assembly.

Reviewer #2 (Remarks to the Author):

I am happy with the responses provided to the reviewers' comments and the additional data included

Reviewer #3 (Remarks to the Author):

The authors have addressed my concerns and recommend for publication.

Response to referee comments

No final comments were given, so we also have no comments to give here.

Susan